# πBO: Augmenting Acquisition Functions with User Beliefs for Bayesian Optimization

**Carl Hvarfner**[1], **Danny Stoll**[2], **Artur Souza**[3], **Marius Lindauer**[4], **Frank Hutter**[2,5] **& Luigi Nardi**[1,6]

[1]Lund University, [2]University of Freiburg, [3]Federal University of Minas Gerais,

[4]Leibniz University Hannover, [5]Bosch Center for Artificial Intelligence, [6]Stanford University

```
{carl.hvarfner, luigi.nardi}@cs.lth.se,
{stolld, fh}@cs.uni-freiburg.de,
arturluis@dcc.ufmg.br, lindauer@tnt.uni-hannover.de
```

## Abstract

Bayesian optimization (BO) has become an established framework and popular tool for hyperparameter optimization (HPO) of machine learning (ML) algorithms. While known for its sample-efficiency, vanilla BO can not utilize readily available prior beliefs the practitioner has on the potential location of the optimum. Thus, BO disregards a valuable source of information, reducing its appeal to ML practitioners. To address this issue, we propose πBO, an acquisition function generalization which incorporates prior beliefs about the location of the optimum in the form of a probability distribution, provided by the user. In contrast to previous approaches, πBO is conceptually simple and can easily be integrated with existing libraries and many acquisition functions. We provide regret bounds when πBO is applied to the common Expected Improvement acquisition function and prove convergence at regular rates independently of the prior. Further, our experiments show that πBO outperforms competing approaches across a wide suite of benchmarks and prior characteristics. We also demonstrate that πBO improves on the state-of-the-art performance for a popular deep learning task, with a $12.5\times$ time-to-accuracy speedup over prominent BO approaches.

## 1 Introduction

The optimization of expensive black-box functions is a prominent task, arising across a wide range of applications. Bayesian optimization (BO) is a sample-efficient approach to cope with this task, and has been successfully applied to various problem settings, including hyperparameter optimization (HPO) (Snoek et al., 2012), neural architecture search (NAS) (Ru et al., 2021), joint NAS and HPO (Zimmer et al., 2021), algorithm configuration (Hutter et al., 2011), hardware design (Nardi et al., 2019), robotics (Calandra et al., 2014), and the game of Go (Chen et al., 2018).

Despite the demonstrated effectiveness of BO for HPO (Bergstra et al., 2011; Turner et al., 2021), its adoption among practitioners remains limited. In a survey covering NeurIPS 2019 and ICLR 2020 (Bouthillier & Varoquaux, 2020), manual search was shown to be the most prevalent tuning method, with BO accounting for less than 7% of all tuning efforts. As the understanding of hyperparameter settings in deep learning (DL) models increase (Smith, 2018), so too does the tuning proficiency of practitioners (Anand et al., 2020). As previously displayed (Smith, 2018; Anand et al., 2020; Souza et al., 2021; Wang et al., 2019), this knowledge manifests in choosing single configurations or regions of hyperparameters that presumably yield good results, demonstrating a belief over the location of the optimum. BO's deficit to properly incorporate said beliefs is a reason why practitioners prefer manual search to BO (Wang et al., 2019), despite its documented shortcomings (Bergstra & Bengio, 2012). To improve the usefulness of automated HPO approaches for ML practitioners, the ability to incorporate such knowledge is pivotal.

Well-established BO frameworks (Snoek et al., 2012; Hutter et al., 2011; The GPyOpt authors, 2016; Kandasamy et al., 2020; Balandat et al., 2020) support user input to a limited extent, such as by biasing the initial design, or by narrowing the search space; however, this type of hard prior can lead to poor performance by missing important regions. BO also supports a prior over functions $p(f)$ via

the Gaussian Process kernel. However, this option for injecting knowledge is not aligned with the knowledge that experts possess: they often know which *ranges of hyperparameter values* tend to work best (Perrone et al., 2019; Smith, 2018; Wang et al., 2019), and are able to specify a probability distribution to quantify these priors. For example, many users of the Adam optimizer (Kingma & Ba, 2015) know that its best learning rate is often in the vicinity of $1 \times 10^{-3}$. In practice, DL experiments are typically conducted in a low-budget setting of less than 50 full model trainings (Bouthillier & Varoquaux, 2020). As such, practitioners want to exploit their knowledge efficiently without wasting early model trainings on configurations they expect to likely perform poorly. Unfortunately, this suits standard BO poorly, as BO requires a moderate number of function evaluations to learn about the response surface and make informed decisions that outperform random search.

While there is a demand to increase knowledge injection possibilities to further the adoption of BO, the concept of encoding prior beliefs over the location of an optimum is still rather novel: while there are some initial works (Ramachandran et al., 2020; Li et al., 2020; Souza et al., 2021), no approach exists so far that allows the integration of arbitrary priors and offers flexibility in the choice of acquisition function; theory is also lacking. We close this gap by introducing a novel, remarkably simple, approach for injecting arbitrary prior beliefs into BO that is easy to implement, agnostic to the surrogate model used and converges at standard BO rates for any choice of prior.

**Our contributions**    After discussing our problem setting, related work, and background (Section 2), we make the following contributions:

1. We introduce $\pi$BO, a novel generalization of myopic acquisition functions that accounts for user-specified prior distributions over possible optima, is demonstrably simple-to-implement, and can be easily combined with arbitrary surrogate models (Section 3.1 & 3.2);

2. We formally prove that $\pi$BO inherits the theoretical properties of the well-established Expected Improvement acquisition function (Section 3.3);

3. We demonstrate on a broad range of established benchmarks and in DL case studies that $\pi$BO can yield $12.5\times$ time-to-accuracy speedup over vanilla BO (Section 4).

## 2    BACKGROUND AND RELATED WORK

### 2.1    BLACK-BOX OPTIMIZATION

We consider the problem of optimizing a black-box function $f$ across a set of feasible inputs $\mathcal{X} \subset \mathbb{R}^d$:

$$\boldsymbol{x}^* \in \underset{\boldsymbol{x} \in \mathcal{X}}{\arg\min} f(\boldsymbol{x}). \tag{1}$$

We assume that $f(x)$ is expensive to evaluate, and can potentially only be observed through a noisy estimate, $y$. In this setting, we wish to minimize $f$ in an efficient manner, typically adhering to a budget which sets a cap on the number of points that can be evaluated.

**Black-Box Optimization with Probabilistic User Beliefs**    In our work, we consider an augmented version of the optimization problem in Eq. (1), where we have access to user beliefs in the form of a probability distribution on the location of the optimum. Formally, we define the problem of black-box optimization with probabilistic user beliefs as solving Eq. (1), given a user-specified prior probability on the location of the optimum defined as

$$\pi(\boldsymbol{x}) = \mathbb{P}\left( f(\boldsymbol{x}) = \min_{\boldsymbol{x}' \in \mathcal{X}} f(\boldsymbol{x}') \right), \tag{2}$$

where regions that the user expects to likely to contain an optimum will have a high value. We note that, without loss of generality, we require $\pi$ to be strictly positive on all of $\mathcal{X}$, i.e., any point in the search space might be an optimum. Since the user belief $\pi(\boldsymbol{x})$ can be inaccurate or even misleading, optimizing Eq. (1) given (2) is a challenging problem.

### 2.2    BAYESIAN OPTIMIZATION

We outline Bayesian optimization (Mockus et al., 1978; Brochu et al., 2010; Shahriari et al., 2016b).

**Model**   BO aims to globally minimize $f$ by an initial experimental design $\mathcal{D}_0 = \{(\boldsymbol{x}_i, y_i)\}_{i=1}^{M}$ and thereafter sequentially deciding on new points $\boldsymbol{x}_n$ to form the data $\mathcal{D}_n = \mathcal{D}_{n-1} \cup \{(\boldsymbol{x}_n, y_n)\}$ for the $n$-th iteration with $n \in \{1 \ldots N\}$. After each new observation, BO constructs a probabilistic surrogate model of $f$ and uses that surrogate to evaluate an acquisition function $\alpha(\boldsymbol{x}, \mathcal{D}_n)$. The combination of surrogate model and acquisition function encodes the policy for selecting the next point $\boldsymbol{x}_{n+1}$. When constructing the surrogate, the most common choice is Gaussian processes (Rasmussen & Williams, 2006), which model $f$ as $p(f|\mathcal{D}_n) = \mathcal{GP}(m, k)$, with prior mean $m$ (which is typically 0) and positive semi-definite covariance kernel $k$. The posterior mean $m_n$ and the variance $s_n^2$ are

$$m_n(\boldsymbol{x}) = \mathbf{k}_n(\boldsymbol{x})^\top (\mathbf{K}_n + \sigma_n^2 \mathbf{I})\mathbf{y}, \quad s_n^2(\boldsymbol{x}) = k(\boldsymbol{x}, \boldsymbol{x}) - \mathbf{k}_n(\boldsymbol{x})^\top (\mathbf{K}_n + \sigma_n^2 \mathbf{I})\mathbf{k}_n(\boldsymbol{x}), \qquad (3)$$

where $(\mathbf{K}_n)_{ij} = k(\boldsymbol{x}_i, \boldsymbol{x}_j)$, $\mathbf{k}_n(\boldsymbol{x}) = [k(\boldsymbol{x}, \boldsymbol{x}_1), \ldots, k(\boldsymbol{x}, \boldsymbol{x}_n)]^\top$ and $\sigma_n^2$ is the estimation of the observation noise variance $\sigma^2$. Alternative surrogate models include Random forests (Hutter et al., 2011) and Bayesian neural networks (Springenberg et al., 2016).

**Acquisition Functions**   To obtain new candidates to evaluate, BO employs a criterion, called an acquisition function, that encapsulates an explore-exploit trade-off. By maximizing this criterion at each iteration, one or more candidate point are obtained and added to observed data. Several acquisition functions are used in BO; the most common of these is Expected Improvement (EI) (Jones et al., 1998). For a noiseless function, EI selects the next point $\boldsymbol{x}_{n+1}$, where $f_n^*$ is the minimal objective function value observed by iteration $n$, as

$$\boldsymbol{x}_{n+1} \in \underset{\boldsymbol{x} \in \mathcal{X}}{\arg\max}\, \mathbb{E}\left[[(f_n^* - f(\boldsymbol{x})]^+\right] = \underset{\boldsymbol{x} \in \mathcal{X}}{\arg\max}\, Z s_n(\boldsymbol{x})\Phi(Z) + s_n(\boldsymbol{x})\phi(Z), \qquad (4)$$

where $Z = (f_n^* - m_n(\boldsymbol{x}))/s_n(\boldsymbol{x})$. Thus, EI provides a myopic strategy for determining promising points; it also comes with convergence guarantees (Bull, 2011). Similar myopic acquisition functions are Upper Confidence Bound (UCB) (Srinivas et al., 2012), Probability of Improvement (PI) (Jones, 2001; Kushner, 1964) and Thompson Sampling (TS) (Thompson, 1933). A different class of acquisition functions is based on non-myopic criteria, such as Entropy Search (Hennig & Schuler, 2012), Predictive Entropy Search (Hernández-Lobato et al., 2014) and Max-value Entropy Search (Wang & Jegelka, 2017), which select points to minimize the uncertainty about the optimum, and the Knowledge Gradient (Frazier et al., 2008), which aims to minimize the posterior mean of the surrogate at the subsequent iteration. Our work applies to all acquisition functions in the first class, and we leave its extension to those in the second class for future work.

## 2.3   RELATED WORK

There are two main categories of approaches that exploit prior knowledge in BO: approaches that use records of previous experiments, and approaches that incorporate assumptions on the black-box function provided either directly or indirectly by the user. As $\pi$BO exploits prior knowledge from users, we briefly discuss approaches which utilize previous experiments, and then comprehensively discuss the literature on exploiting expert knowledge.

**Learning from Previous Experiments**   Transfer learning for BO aims to automatically extract and use knowledge from prior executions of BO. These executions can come, for example, from learning and optimizing the hyperparameters of a machine learning algorithm on different datasets (van Rijn & Hutter, 2018; Swersky et al., 2013; Wistuba et al., 2015; Perrone et al., 2019; Feurer et al., 2015; 2018), or from optimizing the hyperparameters at different development stages (Stoll et al., 2020). For a comprehensive overview of meta learning for hyperparameter optimization, please see the survey from Vanschoren (2018). In contrast to these transfer learning approaches, $\pi$BO and the related work discussed below does not hinge on the existence of previous experiments, and can therefore be applied more generally.

**Incorporating Expert Priors over Function Structure**   BO can leverage structural priors on how the objective function is expected to behave. Traditionally, this is done via the surrogate model's prior over functions, e.g., the kernel of the GP. However, there are lines of work that explore additional structural priors for BO to leverage. For instance, both SMAC (Hutter et al., 2011) and iRace (López-Ibáñez et al., 2016) support structural priors in the form of log-transformations, Li et al. (2018) propose to use knowledge about the monotonicity of the objective function as a prior for BO, and Snoek et al. (2014) model non-stationary covariance between inputs by warping said inputs.

Oh et al. (2018) and Siivola et al. (2018) both propose structural priors tailored to high-dimensional problems, addressing the issue of over-exploring the boundary described by Swersky (2017). Oh et al. (2018) propose a cylindrical kernel that expands the center of the search space and shrinks the edges, while Siivola et al. (2018) propose adding derivative signs to the edges of the search space to steer BO towards the center. Lastly, Shahriari et al. (2016a) propose a BO algorithm for unbounded search spaces which uses a regularizer to penalize points based on their distance to the center of the user-defined search space. All of these approaches incorporate prior information on specific properties of the function or search space, and are thus not always applicable. Moreover, they do not generally direct the search to desired regions of the search space, offering the user little control over the selection of points to evaluate.

**Incorporating Expert Priors over Function Optimum** Few previous works have proposed to inject explicit prior distributions over the location of an optimum into BO. In these cases, users explicitly define a prior that encodes their beliefs on where the optimum is more likely to be located. Bergstra et al. (2011) suggest an approach that supports prior beliefs from a fixed set of distributions. However, this approach cannot be combined with standard acquisition functions. BOPrO (Souza et al., 2021) employs a similar structure that combines the user-provided prior distribution with a data-driven model into a pseudo-posterior. From the pseudo-posterior, configurations are selected using the EI acquisition function, using the formulation in Bergstra et al. (2011). While BOPrO is able to recover from misleading priors, its design restricts it to only use EI. Moreover, it does not provide the convergence guarantees of $\pi$BO.

Li et al. (2020) propose to infer a posterior conditioned on both the observed data and the user prior through repeated Thompson sampling and maximization under the prior. This method displays robustness against misleading priors but lacks in empirical performance. Additionally, it is restricted to only one specific acquisition function. Ramachandran et al. (2020) use the probability integral transform to warp the search space, stretching high-probability regions and shrinking others. While the approach is model- and acquisition function agnostic, it requires invertible priors, and does not empirically display the ability to recover from misleading priors. In Section 4, we demonstrate that $\pi$BO compares favorably for priors over the function optimum, and shows improved empirical performance. Additionally, we do a complete comparison of all approaches in Appendix C.

In summary, $\pi$BO sets itself apart from the methods above by being simpler (and thus easier to implement in different frameworks), flexible with regard to different acquisition functions and different surrogate models, the availability of theoretical guarantees, and, as we demonstrate in Section 4, better empirical results.

## 3 METHODOLOGY

We now present $\pi$BO, which allows users to specify their belief about the location of the optimum through any probability distribution. A conceptually simple approach, $\pi$BO can be easily implemented in existing BO frameworks and can be combined directly with the myopic acquisition functions listed above. $\pi$BO augments an acquisition function to emphasize promising regions under the prior, ensuring such regions are to be explored frequently. As optimization progresses, the $\pi$BO strategy increasingly resembles that of vanilla BO, retaining its standard convergence rates (see Section 3.3). $\pi$BO is publicly available as part of the SMAC (`https://github.com/automl/SMAC3`) and HyperMapper (`https://github.com/luinardi/hypermapper`) HPO frameworks.

### 3.1 PRIOR-WEIGHTED ACQUISITION FUNCTION

In $\pi$BO, we consider $\pi(\boldsymbol{x})$ in Eq. (2) to be a weighting scheme on points in $\mathcal{X}$. The heuristic provided by an acquisition function $\alpha(\boldsymbol{x}, \mathcal{D}_n)$, such as EI in Eq. (2.2), can then be combined with said weighting scheme to form a prior-weighted version of the acquisition function. The resulting strategy then becomes:

$$\boldsymbol{x}_n \in \operatorname*{arg\,max}_{\boldsymbol{x} \in \mathcal{X}} \alpha(\boldsymbol{x}, \mathcal{D}_n)\pi(\boldsymbol{x}). \tag{5}$$

This emphasizes good points under $\pi(\boldsymbol{x})$ throughout the optimization. While this property is suitable for well-located priors $\pi$, it risks incurring a substantial slowdown for poorly-chosen priors; we will now show how to counter this by decaying the prior over time.

## 3.2 Decaying Prior-weighted Acquisition Function

As the optimization progresses, we should increasingly trust the surrogate model over the prior; the model improves with data while the user prior remains fixed. This cannot be achieved with the formulation in Eq. (5), as poorly-chosen priors would permanently slow down the optimization. Rather, to accomplish this desired behaviour, the influence of the prior needs to decay over time. Building on the approaches of Lee et al. (2020) and Souza et al. (2021), we accomplish this by raising the prior to a power $\gamma_n \in \mathbb{R}^+$, which decays towards zero with growing $n$. Thus, the resulting prior $\pi_n(\boldsymbol{x}) = \pi(\boldsymbol{x})^{\gamma_n}$ reflects a belief on the location of an optimum that gets weaker with time, converging towards a uniform distribution. We set $\gamma_n = \beta/n$, where $\beta \in \mathbb{R}^+$ is a hyperparameter set by the user, reflecting their confidence in $\pi(\boldsymbol{x})$. We provide a sensitivity study on $\beta$ in Appendix A. For a given acquisition function $\alpha(\boldsymbol{x}, \mathcal{D}_n)$ and user-specified prior $\pi(\boldsymbol{x})$, we define the decaying prior-weighted acquisition function at iteration $n$ as

$$\alpha_{\pi,n}(\boldsymbol{x}, \mathcal{D}_n) \triangleq \alpha(\boldsymbol{x}, \mathcal{D}_n)\pi_n(\boldsymbol{x}) \triangleq \alpha(\boldsymbol{x}, \mathcal{D}_n)\pi(\boldsymbol{x})^{\beta/n} \tag{6}$$

and its accompanying strategy as the maximizer of $\alpha_{\pi,n}$. With the acquisition function in Eq. (6), the prior will assume large importance initially, promoting the selection of points close to the prior mode. With time, the exponent on the prior will tend to zero, making the prior tend to uniform. Thus, with increasing $n$, the point selection of $\alpha_{\pi,n}$ becomes increasingly similar to that of $\alpha$. Algorithm 1 displays the simplicity of the new strategy, highlighting the required one-line change (Line 6) in the main BO loop. In Line 3, the mode of the prior is used as a first initial sample if available. Otherwise, only sampling is used for initialization.

---

**Algorithm 1** $\pi$BO Algorithm

1: **Input:** Input space $\mathcal{X}$, prior distribution over optimum $\pi(\boldsymbol{x})$, prior confidence parameter $\beta$, size $M$ of the initial design, max number of optimization iterations $N$.
2: **Output:** Optimized design $\boldsymbol{x}^*$.
3: $\{\boldsymbol{x}_i\}_{i=1}^M \sim \pi(\boldsymbol{x})$, $\{y_i \leftarrow f(\boldsymbol{x}_i) + \epsilon_i\}_{i=1}^M$, $\quad \epsilon_i \sim N(0, \sigma^2)$
4: $\mathcal{D}_0 \leftarrow \{(\boldsymbol{x}_i, y_i)\}_{i=1}^M$
5: **for** $\{n = 1, 2, \ldots, N\}$ **do**
6: $\quad \boldsymbol{x}_{new} \leftarrow \arg\max_{\boldsymbol{x} \in \mathcal{X}} \alpha(\boldsymbol{x}, \mathcal{D}_{n-1})\pi(\boldsymbol{x})^{\beta/n}$
7: $\quad y_{new} \leftarrow f(\boldsymbol{x}_{new}) + \epsilon_i$
8: $\quad \mathcal{D}_n \leftarrow \mathcal{D}_{n-1} \cup \{(\boldsymbol{x}_{new}, y_{new})\}$
9: **end for**
10: **return** $\boldsymbol{x}^* \leftarrow \arg\min_{(\boldsymbol{x}_i, y_i) \in \mathcal{D}_N} y_i$

---

To illustrate the behaviour of $\pi$BO, we consider a toy problem with Gaussian priors on three different locations of the 1D space (center, left and right) as displayed in Figure 1. We define a 1D-Log-Branin toy problem by setting the second dimension of the 2D Branin function to the global optimum $x_2 = 2.275$ and optimizing for the first dimension. Initially (iteration 4 in the top row), $\pi$BO amplifies the acquisition function $\alpha$ in high-probability regions, putting a lot of trust in the prior. As the prior decays (iteration 6 and 8 in the middle and bottom rows, respectively), the influence of the prior on the point selection decreases. By later iterations, $\pi$BO has searched substantially around the prior mode, and moves gradually towards other parts of the search space. This is of particular importance for the scenarios in the right column, where $\pi$BO recovers from a misleading prior. In Appendix B, we show that $\pi$BO is applicable to different surrogate models and acquisition functions.

## 3.3 Theoretical Analysis

We now study the $\pi$BO method from a theoretical standpoint when paired with the EI acquisition function. For the full proof, we refer the reader to Appendix E. To provide convergence rates, we rely on the set of assumptions introduced by Bull (2011). These assumptions are satisfied for popular kernels like the Matérn (1960) class and the Gaussian kernel, which is obtained in the limit $\nu \to \infty$, where the rate $\nu$ controls the smoothness of functions from the GP prior. Our theoretical results apply when both length scales $\ell$ and the global scale of variation $\sigma$ are fixed; these results can then be extended to the case where the kernel hyperparameters are learned using Maximum Likelihood

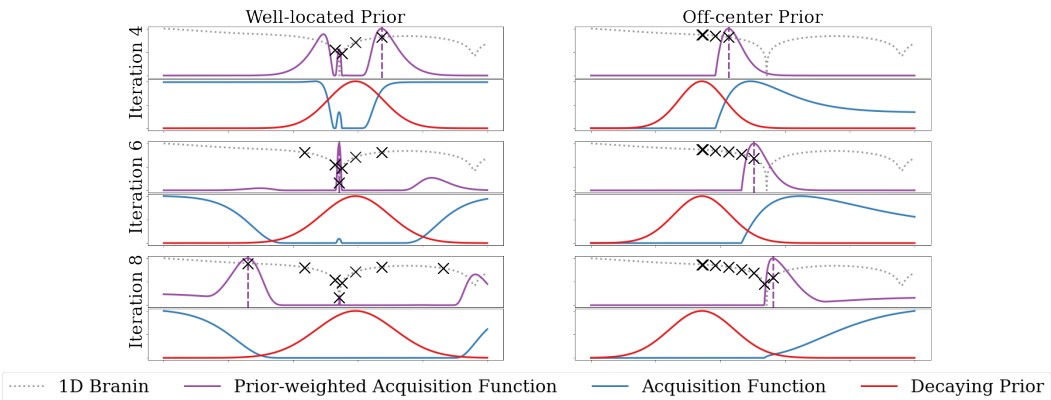

Figure 1: Rescaled values of prior-weighted EI (purple), EI (blue) and $\pi_n$ (red) on a 1D-Branin in logscale (grey) with global optimum in the center of the search space. Runs with two different prior locations ("Well-located" slightly right of optimum, "Off-center" significantly left of optimum) are shown in the two columns. Each row represents an iteration (iteration 4, 6 and 8), for an optimization run with $\beta = 2$. The current selection can be seen as a vertical violet line, and all previous observations are marked as crosses. $\pi$BO amplifies EI in a gradually increasing region around the prior, and moves away from the prior as iterations progress. This is particularly evident in the Off-center example.

Estimation (MLE) following the same procedure as in Bull (2011) (Theorem 5). We define the loss over the ball $B_R$ for a function $f$ of norm $||f||_{\mathcal{H}_{\boldsymbol{\ell}}(\mathcal{X})} \leq R$ in the reproducing kernel Hilbert space (RKHS) $\mathcal{H}_{\boldsymbol{\ell}}(\mathcal{X})$ given a symmetric positive-definite kernel $K_{\boldsymbol{\ell}}$ as

$$\mathcal{L}_n(u, \mathcal{D}_n, \mathcal{H}_{\boldsymbol{\ell}}(\mathcal{X}), R) \triangleq \sup_{||f||_{\mathcal{H}_{\boldsymbol{\ell}}(\mathcal{X})} \leq R} \mathbb{E}_f^u[f(\boldsymbol{x}_n^*) - \min f], \tag{7}$$

where $n$ is the optimization iteration and $u$ a strategy. We focus on the strategy that maximizes $\text{EI}_\pi$, the prior-weighted EI, and show that the loss in Equation (7) can, at any iteration $n$, be bounded by the vanilla EI loss function. We refer to $\text{EI}_{\pi,n}$ and $\text{EI}_n$ when we want to emphasize the iteration $n$ for the acquisition functions $\text{EI}_\pi$ and EI, respectively.

**Theorem 1.** *Given $\mathcal{D}_n$, $K_{\boldsymbol{\ell}}$, $\pi$, $\beta$, $\sigma$, $\boldsymbol{\ell}$, $R$ and the compact set $\mathcal{X} \subset \mathbb{R}^d$ as defined above, the loss $\mathcal{L}_n$ incurred at iteration $n$ by $EI_{\pi,n}$ can be bounded from above as*

$$\mathcal{L}_n(EI_{\pi,n}, \mathcal{D}_n, \mathcal{H}_{\boldsymbol{\ell}}(\mathcal{X}), R) \leq C_{\pi,n}\mathcal{L}_n(EI_n, \mathcal{D}_n, \mathcal{H}_{\boldsymbol{\ell}}(\mathcal{X}), R), \quad C_{\pi,n} = \left(\frac{\max_{\boldsymbol{x} \in \mathcal{X}} \pi(\boldsymbol{x})}{\min_{\boldsymbol{x} \in \mathcal{X}} \pi(\boldsymbol{x})}\right)^{\beta/n}. \tag{8}$$

Using Theorem 1, we obtain the convergence rate of $\text{EI}_\pi$. This trivially follows when considering the fraction of the losses in the limit and inserting the original convergence rate on EI as in Bull (2011):

**Corollary 1.** *The loss of a decaying prior-weighted Expected Improvement strategy, $EI_\pi$, is asymptotically equal to the loss of an Expected Improvement strategy, EI:*

$$\mathcal{L}_n(EI_{\pi,n}, \mathcal{D}_n, \mathcal{H}_{\boldsymbol{\ell}}(\mathcal{X}), R) \sim \mathcal{L}_n(EI_n, \mathcal{D}_n, \mathcal{H}_{\boldsymbol{\ell}}(\mathcal{X}), R), \tag{9}$$

*so we obtain a convergence rate for $EI_\pi$ of $\mathcal{L}_n(EI_{\pi,n}, \mathcal{D}_n, \mathcal{H}_{\boldsymbol{\ell}}(\mathcal{X}), R) = \mathcal{O}(n^{-(\nu \wedge 1)/d}(\log n)^\gamma)$.*

Thus, we determine that the weighting introduced by $\text{EI}_\pi$ does not negatively impact the worst-case convergence rate. The short-term performance is controlled by the user in their choice of $\pi(\boldsymbol{x})$ and $\beta$. This result is coherent with intuition, as a weaker prior or quicker decay will yield a short-term performance closer to that of EI. In contrast, a stronger prior or slower decay does not guarantee the same short-term performance, but can produce better empirical results, as shown in Section 4.

## 4 RESULTS

We empirically demonstrate the efficiency of $\pi$BO in three different settings. As $\pi$BO is a general method to augment acquisition functions, it can be implemented in different parent BO packages, and the implementation in any given package inherits the pros and cons of that package. To minimize confounding factors concerning this choice of parent package, we keep comparisons within the methods in one package where possible and provide results in the other packages in Appendix C. In Sec. 4.2, using Spearmint as a parent package, we evaluate $\pi$BO against three intuitive baselines to assess its performance and robustness on priors with different qualities, ranging from very accurate to purposefully detrimental. To this end, we use toy functions and cheap surrogates, where priors of known quality can be obtained. Next, in Sec. 4.3, we compare $\pi$BO against two competitive approaches (BOPrO and BOWS) that integrate priors over the optimum similarly to $\pi$BO, using HyperMapper (Nardi et al., 2019) as a parent framework, in which the most competitive baseline BOPrO is implemented. For these experiments we adopt a Multilayer Perceptron (MLP) benchmark on various datasets, using the interface provided by HPOBench (Eggensperger et al., 2021), with priors constructed around the defaults provided by the library. Lastly, in Sec. 4.4, we apply $\pi$BO and other approaches to two deep learning tasks, also using priors derived from publicly available defaults. Further, we demonstrate the flexibility of $\pi$BO in Appendix B, where we evaluate $\pi$BO in SMAC (Hutter et al., 2011; Lindauer et al., 2021) with random forests, as another framework with another surrogate model, and adapt it to use the UCB, TS and PI acquisition functions instead of EI.

### 4.1 EXPERIMENTAL SETUP

**Priors**   For our surrogate and toy function tasks, we follow the prior construction methodology in BOPrO (Souza et al., 2021) and create three main types of prior qualities, all Gaussian: strong, weak and wrong. The strong and weak priors are located to have a high and moderate density on the optimum, respectively. The wrong prior is a narrow distribution located in the worst region of the search space. For the OpenML MLP tuning benchmark, we utilize the defaults and search spaces provided in HPOBench (Eggensperger et al., 2021), and construct Gaussian priors for each hyperparameter with their mean on the default value, and a standard deviation of 25% of the hyperparameter's domain. For the DL case studies, we utilize defaults from each task's repository and, for numerical hyperparameters, once again set the standard deviation to 25% of the hyperparameter's domain. For categorical hyperparameters, we place a higher probability on the default. As such, the quality of the prior is ultimately unknown, but serves as a proxy for what a practitioner may choose and has shown to be a reasonable choice (Anastacio & Hoos, 2020). For all experiments, we run $\pi$BO with $\beta = N/10$, where $N$ is the total number of iterations, in order to make the prior influence approximately equal in all experiments, regardless of the number of allowed BO iterations. We investigate the sensitivity to $\beta$ in Appendix A, and the sensitivity to prior quality in Appendix G.

**Baselines**   We empirically evaluate $\pi$BO against the most competitive approaches for priors over the optimum described in Section 2.3: BOPrO (Souza et al., 2021) and BO in Warped Space (BOWS) (Ramachandran et al., 2020). To contextualize the performance of $\pi$BO, we provide additional, simpler baselines: random sampling, sampling from the prior and BO with prior-based initial design. The latter is initialized with the mode of the prior *in addition* to its regular initial design. In our main results, we choose Spearmint (with EI) (Snoek et al., 2012) for this mode-initialized baseline, simply referring to it as Spearmint. See Appendix F for complete details on the experiments.

### 4.2 ROBUSTNESS OF $\pi$BO

First, we study the robustness of $\pi$BO. To this end, we show that $\pi$BO benefits from informative priors and can recover from wrong priors, being consistent with our theoretical results in Section 3.3. To this end, we consider a well-known black-box optimization function, Branin (2D), as well as two surrogate HPO tasks from the Profet suite (Klein et al., 2019): FC-Net (6D) and XGBoost (8D). For these tasks, we exemplarily show results for $\pi$BO implemented in the Spearmint framework. As Figure 2 shows, $\pi$BO is able to quickly improve over sampling from the prior. Moreover, it improves substantially over Spearmint (with mode initialization) for all informative priors, staying up to an order of magnitude ahead throughout the optimization for both strong and weak priors. For wrong priors, $\pi$BO displays desired robustness by recovering to approximately equal regret as Spearmint. In contrast, Spearmint frequently fails to substantially improve from its initial design on the strong and

weak prior, which demonstrates the importance of considering the prior throughout the optimization procedure. This effect is even more pronounced on the higher-dimensional tasks FCNet and XGBoost, where BO typically spends many iterations at the boundary (Swersky, 2017). Here, $\pi$BO rapidly improves multiple orders of magnitude over the initial design, displaying its ability to efficiently exploit the information provided by the prior.

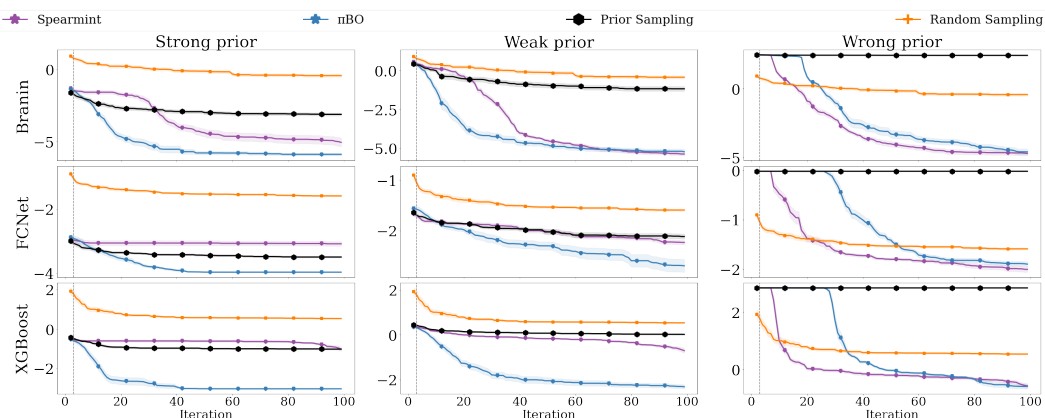

Figure 2: Comparison of $\pi$BO, Spearmint, and two sampling approaches on Branin, FCNet and XGBoost for various prior strengths. Mean and standard error of log simple regret is displayed over 100 iterations, averaged over 20 runs. The vertical line represents the end of the initial design phase.

### 4.3 COMPARISON OF $\pi$BO AGAINST OTHER PRIOR-GUIDED APPROACHES

Next, we study the performance of $\pi$BO against other state-of-the-art prior-guided approaches. To this end, we consider optimizing 5 hyperparameters of an MLP for classification (Eggensperger et al., 2021) on 6 different OpenML datasets (Vanschoren et al., 2014) and compare against BOPrO (Souza et al., 2021) and BOWS (Ramachandran et al., 2020). For minimizing confounding factors, we implement $\pi$BO and BOWS in HyperMapper (Nardi et al., 2019), the same framework that BOPrO runs on. Moreover, we let all approaches share $\pi$BO's initialization procedure. We consider a budget of 50 iterations as it is common with ML practitioners (Bouthillier & Varoquaux, 2020). In Figure 3, we see that $\pi$BO offers the best performance on four out of six tasks, and displays the most consistent performance across tasks. In contrast to them BOWS and BOPrO, $\pi$BO also comes with theoretical guarantees and is flexible in the choice of framework and acquisition function.

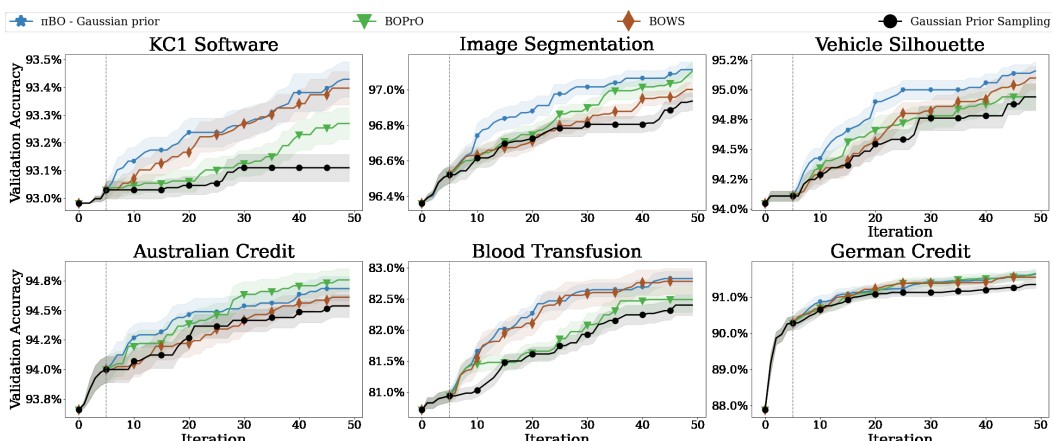

Figure 3: Comparison of $\pi$BO, BOPrO, BOWS, and prior sampling for 5D MLP tuning on various OpenML datasets for a prior centered on default values. We show mean and standard error of the accuracy across 20 runs. The vertical line represents the end of the initial design phase.

## 4.4 Case Studies on Deep Learning Pipelines

Last, we study the impact of $\pi$BO on deep learning applications, which are often fairly expensive, making efficiency even more important than in HPO for traditional machine learning. To this end, we consider two deep learning case studies: segmentation of neuronal processes in electron microscopy images with a U-Net(6D) (Ronneberger et al., 2015), with code provided from the NVIDIA deep learning examples repository (Przemek et al.), and image classification on ImageNette-128 (6D) (Howard, 2019), a light-weight adaptation of ImageNet (Deng et al., 2009), with code from the repository of the popular FastAI library (Howard et al., 2018). We mimic the setup from Section 4.3 by using the HyperMapper framework and identical initialization procedures across approaches. Gaussian priors are set on publicly available default values, which are results of previous tuning efforts of the original authors. We again optimize for a practical budget of 50 iterations (Bouthillier & Varoquaux, 2020). As test splits for both tasks were not available to us, we report validation scores.

As shown in Figure 4, $\pi$BO achieves a $2.5\times$ time-to-accuracy speedup over Vanilla BO. For ImageNette, the performance of $\pi$BO at iteration 4 already surpasses the performance of Vanilla BO at Iteration 50, demonstrating a $12.5\times$ time-to-accuracy speedup. Ultimately, $\pi$BO's final performance establishes a new state-of-the-art validation performance on ImageNette with the provided pipeline, with a final accuracy of 94.14% (vs. the previous state of the art with 93.55%[1]).

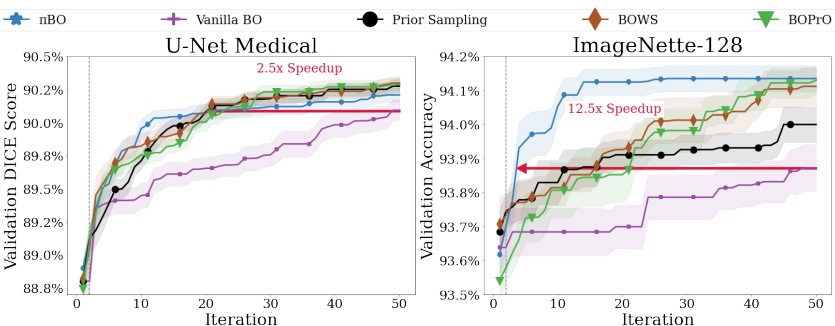

Figure 4: Comparison of approaches for U-Net Medical and ImageNette-128 for a prior centered on default values. We show mean and standard error of the accuracy across 20 runs for U-Net Medical and 10 runs for ImageNette-128. The vertical line represents the end of the initial design phase.

## 5 Conclusion and Future Work

We presented $\pi$BO, a conceptually very simple Bayesian optimization approach for leveraging user beliefs about the location of an optimum, which relies on a generalization of myopic acquisition functions. $\pi$BO modifies the selection of design points through a decaying weighting scheme, promoting high-probability regions under the prior. Contrary to previous approaches, $\pi$BO imposes only minor restrictions on the type of priors, surrogates or frameworks that can be used. $\pi$BO provably converges at regular rates, displays state-of-the art performance across tasks, and effectively recovers from poorly specified priors. Moreover, we have demonstrated that $\pi$BO can yield substantial performance gains for practical low-budget settings, improving on the state-of-the-art for a real-world CNN tuning tasks even with trivial choices for the prior. For practitioners who have historically relied on manual or grid search for HPO, we hope that $\pi$BO will serve as an intuitive and effective tool for bridging the gap between traditional tuning methods and BO.

$\pi$BO sets the stage for several follow-up studies. Amongst others, we will examine the extension of $\pi$BO to non-myopic acquisition functions, such as entropy-based methods. Non-myopic acquisition functions do not fit well in the current $\pi$BO framework, as they do not necessarily benefit from evaluating inputs expected to perform well. We will also combine $\pi$BO with multi-fidelity optimization methods to yield even higher speedups, and with multi-objective optimization to jointly optimize performance and secondary objective functions, such as interpretability or fairness of models.

---

[1]`https://github.com/fastai/imagenette#imagenette-leaderboard`, 80 Epochs, 128 Resolution

## 6 ETHICS STATEMENT

Our work proposes an acquisition function generalization which incorporates prior beliefs about the location of the optimum into optimization. The approach is foundational and thus will not bring direct societal or ethical consequences. However, $\pi$BO will likely be used in the development of applications for a wide range of areas and thus indirectly contribute to their impacts on society. In particular, we envision that $\pi$BO will impact a multitude of fields by allowing ML experts to inject their knowledge about the location of the optimum into Bayesian Optimization.

We also note that we intend for $\pi$BO to be a tool that allows users to assist Bayesian Optimization by providing reasonable prior knowledge and beliefs. This process induces user bias into the optimization, as $\pi$BO will inevitably start by optimizing around this prior. As some users may only be interested in optimizing in the direct neighborhood of their prior, $\pi$BO could allow them to do so if provided with a high $\beta$ value in relation to the number of iterations. Thus, if improperly specified, $\pi$BO could serve to reinforce user's beliefs by providing improved solutions only for the user's region of interest. However, if used properly, $\pi$BO will reduce the computational resources required to find strong hyperparameter settings, contributing to the sustainability of machine learning.

## 7 REPRODUCIBILITY

In order to make the experiments run in $\pi$BO as reproducible as possible, we have included links to repositories of our implementations in both Spearmint and HyperMapper, with instructions on how to run our experiments. Moreover, we have included in said repositories all of the exact priors that we have used for our runs, which run out of the box. The priors we used were, in our opinion, well motivated as to avoid subjectivity, which we hope serves as a good frame of reference for similar works in the future. Specifically, Appendix 4.4 describes how we ran our DL experiments, Appendix F.1 goes into the implementation in further detail, and Appendix D displays the exact priors for all our experiments and prior strengths. Our Spearmint implementation of both $\pi$BO and BOWS is available at `https://github.com/piboauthors/PiBO-Spearmint`, and our HyperMapper implementation is available at `https://github.com/piboauthors/PiBO-Hypermapper`. For our results on the convergence of $\pi$BO, we have provided a complete proof in Appendix E.

## 8 ACKNOWLEDGEMENTS

Luigi Nardi was supported in part by affiliate members and other supporters of the Stanford DAWN project — Ant Financial, Facebook, Google, Intel, Microsoft, NEC, SAP, Teradata, and VMware. Carl Hvarfner and Luigi Nardi were partially supported by the Wallenberg AI, Autonomous Systems and Software Program (WASP) funded by the Knut and Alice Wallenberg Foundation. Artur Souza was supported by CAPES, CNPq, and FAPEMIG. Frank Hutter acknowledges support by the European Research Council (ERC) under the European Union Horizon 2020 research and innovation programme through grant no. 716721, through TAILOR, a project funded by the EU Horizon 2020 research and innovation programme under GA No 952215, by the Deutsche Forschungsgemeinschaft (DFG, German Research Foundation) under grant number 417962828 and by the state of Baden-Württemberg through bwHPC and the German Research Foundation (DFG) through grant no INST 39/963-1 FUGG. Marius Lindauer acknowledges support by the European Research Council (ERC) under the Europe Horizon programme. The computations were also enabled by resources provided by the Swedish National Infrastructure for Computing (SNIC) at LUNARC partially funded by the Swedish Research Council through grant agreement no. 2018-05973.

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

# A    BETA ABLATION STUDY

We consider the effect of the $\beta$ hyperparameter of $\pi$BO introduced in Section 3.2, controlling the speed of the prior decay. To show the effect of this hyperparameter, we display the performance of $\pi$BO for the toy and surrogate-based benchmarks across all prior qualities. We emphasize the trade-off between high-end performance on good priors and robustness to bad priors. In general, a higher value of $\beta$ yields better performance for good priors, but makes $\pi$BO slower to recover from bad priors. This behaviour follows intuition and the results provided in Section 3.3.

In Figure 5, we display how $\pi$BO performs for different choices of $\beta$, and once again provide sampling from the prior and Spearmint as baselines. Following the prior decay parameter baseline by (Souza et al., 2021), we show that the choice of $\beta = 10$ onsistently gives one of the best performances for strong priors, while retaining good overall robustness. Nearly all choices of $\beta$ give a final performance better than that of Spearmint for good priors. Additionally, there is a clear relationship between final performance and $\beta$ on all good priors. This is best visualized in the weak XGBoost experiment, where the final performances are distinctly sorted by increasing $\beta$. Similar patterns are not as apparent in the final performance on wrong priors. This behaviour highlights the benefits of slowly decaying the prior. Overall, $\pi$BO is competitive for a wide range of $\beta$, but suffers slightly worse final performance on good priors for low values of $\beta$.

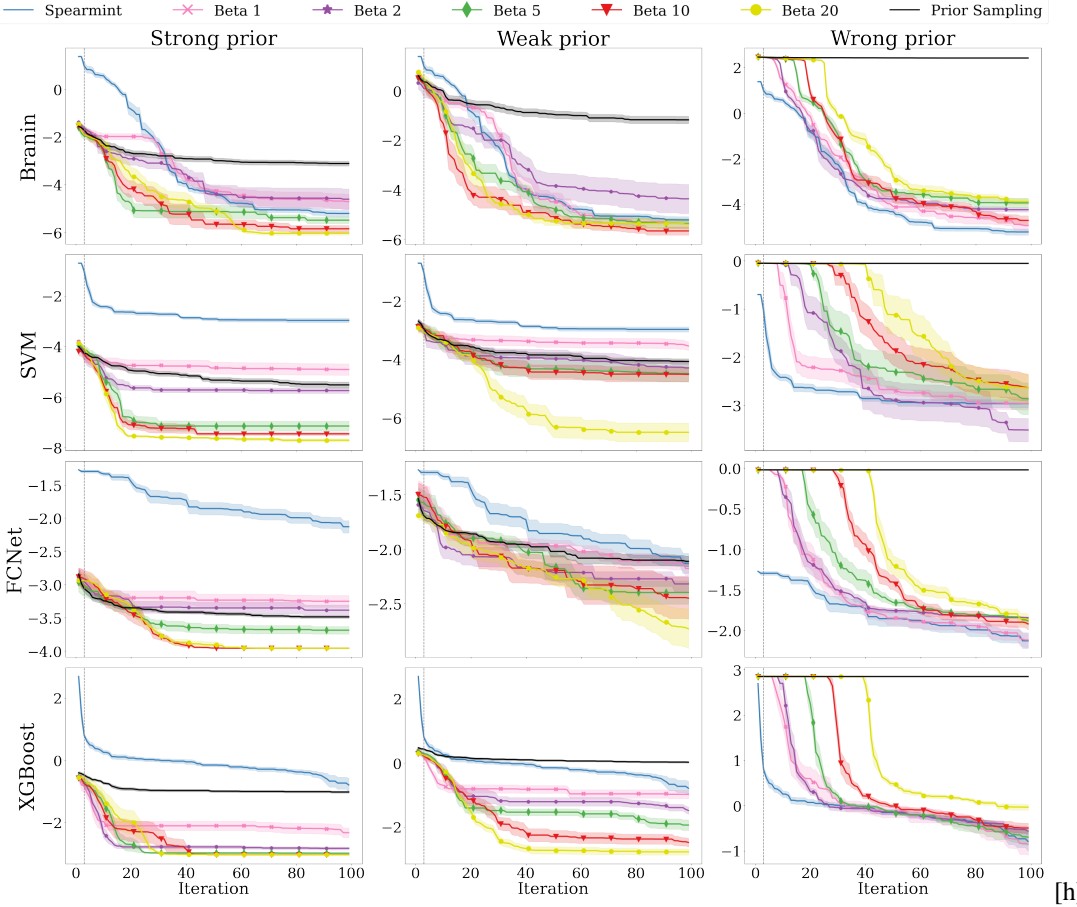

Figure 5: Comparison of $\pi$BO and Spearmint with varying values of $\beta$ for Branin and Profet benchmarks for the strong, weak and wrong prior qualities. The mean and standard error of log simple regret is displayed over 100 iterations, averaged over 10 repetitions. Iteration 1 is removed for visibility purposes. The vertical line represents the end of the initial design phase.

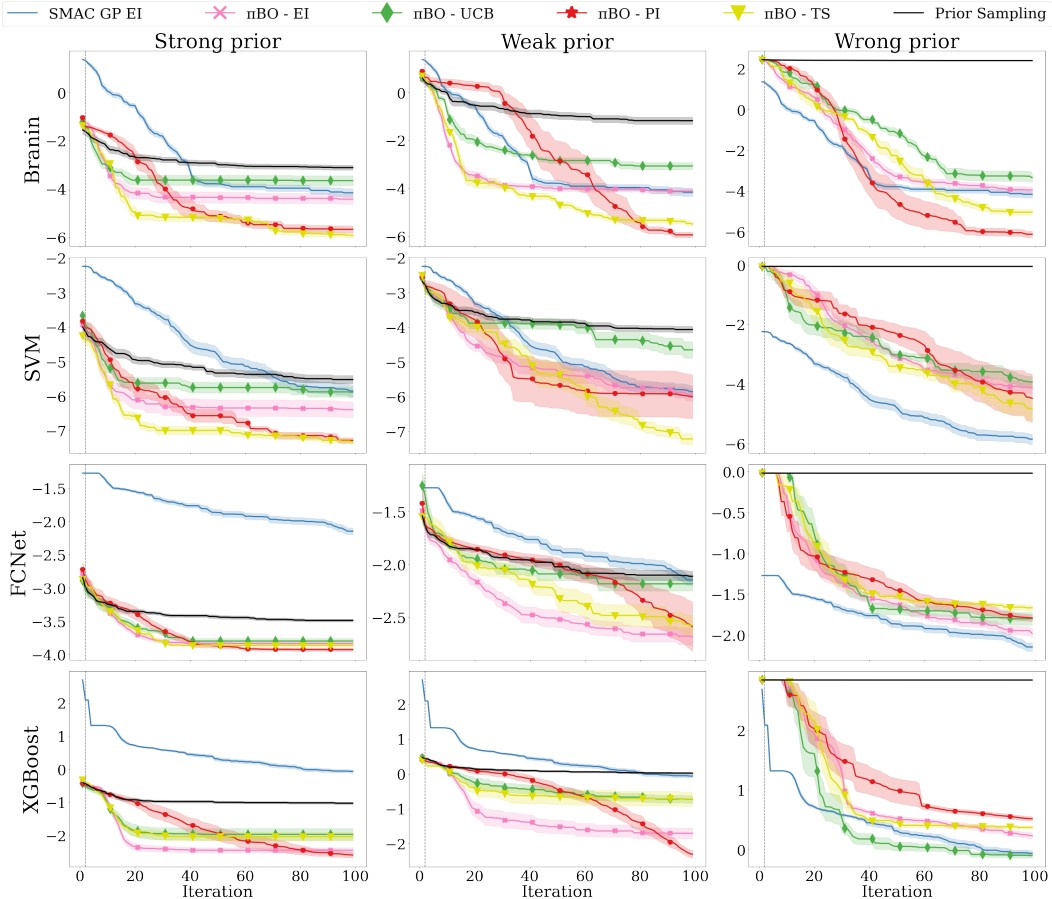

Figure 6: $\pi$BO on SMAC using a GP surrogate for the EI, UCB, PI and TS acquisition functions, $\beta = 10$ for Branin and Profet benchmarks for varying prior qualities. The mean and standard error of log simple regret is displayed over 100 iterations, averaged over 10 repetitions. Iteration 1 is removed for visibility purposes. The vertical line represents the end of the initial design phase.

## B  $\pi$BO VERSATILITY

We show the versatility of $\pi$BO by implementing it in numerous variants of SMAC Hutter et al. (2011), a well-established HPO framework which supports both GP and RF surrogates, and a majority of the myopic acquisition functions mentioned in Section 2. We showcase the performance of $\pi$BO-EI, $\pi$BO-PI, $\pi$BO-UCB and $\pi$BO-TS on the general formulation of $\pi$BO with a GP surrogate, as well as $\pi$BO-EI with an RF surrogate, which requires a minor adaptation.

### B.1  GENERAL FORMULATION OF $\pi$BO

To allow for the universality of $\pi$BO across several acquisition function, we must consider the various magnitudes of acquisition functions. As UCB and TS typically output values in the same order of magnitude and sign as the objective function, we do not want the behaviour of $\pi$BO to be affected by such variations.

The solution to the problem referenced above is to add a simple affine transformation to the observations, $\{y_i\}_{i=1}^n$, by subtracting by the incumbent, $y_n^*$. As such, we consider at each time step not the original dataset, $\mathcal{D}_n = \{(\boldsymbol{x}_i, y_i)\}_{i=1}^n$, but the augmented dataset $\hat{\mathcal{D}}_n = \{(\boldsymbol{x}_i, y_i - y_n^*)\}_{i=1}^n$. With this formulation, we get the desired scale- and sign-invariance in the UCB and TS acquisition functions, without changing the original strategy of any of the acquisition function. Notably, this change leaves prior-weighted EI and PI unaffected.

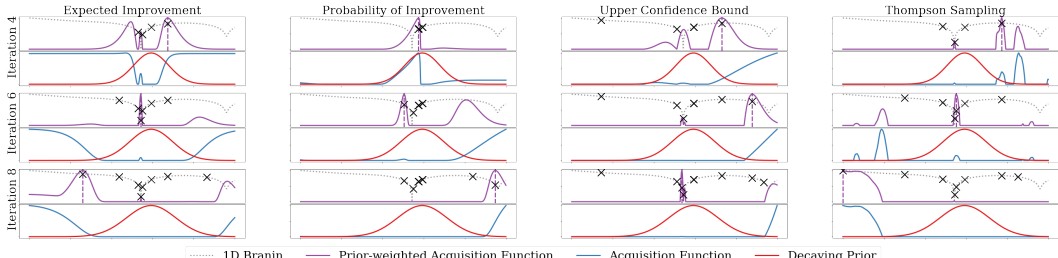

Figure 7: Point selection strategy evolution for $\pi$BO on SMAC with a GP surrogate for the EI, PI, UCB and TS acquisition functions on the Log-1D-Branin function. The plots show rescaled values of the prior-weighted acquisition function (purple), the regular acquisition function (blue) and $\pi_n$ (red) on a 1D-Branin (grey) with global optimum in the center of the search space and the current selection as a vertical violet line. The rows represent iterations 4, 6 and 8 for a run with $\beta = 2$.

## B.2 RANDOM FOREST SURROGATE

We now demonstrate $\pi$BO with a RF surrogate model. In the SMAC implementation of the RF surrogate, the model forms piece-wise constant mean and covariance functions. Naturally, this leads to the EI, PI or UCB acquisition function surface being piece-wise constant as well. Consequently, an acquisition function with a RF surrogate will typically have a region of global optima. The choice of the next design point is then selected uniformly at random among the candidate optima. We wish to retain this randomness when applying $\pi$BO. As such, we require the prior to be piece-wise constant, too. To do so, we employ a binning approach, that linearly rounds prior values after applying the decay term. The granularity of the binning decreases at the same rate as the prior, allowing the piece-wise constant regions of the prior grow in size as optimization progresses. In Figure 9, we demonstrate the importance of the piece-wise constant acquisition function by showing the point selection when applying a $\pi$BO with a continuous prior to an RF surrogate (left) and when applying the binning approach (right). Notably, the smooth prior on the left repeatedly proposes design points very close to previous points, as the prior forces the selection of points near the boundary of a promising region. Thus, the surrogate model rarely improves, and the optimization gets stuck at said boundary for multiple iterations at a time. This is best visualized at iteration 5 and 10, where similar points have been selected for all iterations in the time span. With the binned prior on the right, the selection of design points occurs randomly within a region, avoiding the static point selection and updating of non-modified approach. In Figure 8, we report the performance of $\pi$BO with a RF surrogate and the binning approach. This approach is competitive, as it provides substantial improvement over SMAC, improves over sampling from the prior, and quickly recovers from misleading priors. Notably, the binning is not required for discrete parameters, as the piece-wise constant property holds by default. Thus, this adaptation is only necessary for continuous parameters.

## C OTHER PRIOR-BASED APPROACHES

We now demonstrate the performance of $\pi$BO for five different functions and HPO Surrogates: Branin, Hartmann-6, as well as three tasks from the Profet suite - SVM, FCNet and XGBoost. We compare all frameworks for priors over the optimum - namely BOPrO Souza et al. (2021), BOWS Ramachandran et al. (2020), TPE Bergstra et al. (2011), PS-G Li et al. (2020). The performance of $\pi$BO is shown on two different frameworks - Spearmint and Hypermapper - to allow for fair comparison and display cross-framework consistency. As BOWS is implemented in Spearmint and BOPrO in Hypermapper, they appear in the plots retaining to their framework. We display each approach with vanilla Spearmint/Hypermapper, with normal initialization, as an additional baseline. Moreover, we display the performance of $\pi$BO implemented in Spearmint, as well as Mode + Spearmint, on the MLP tuning tasks.

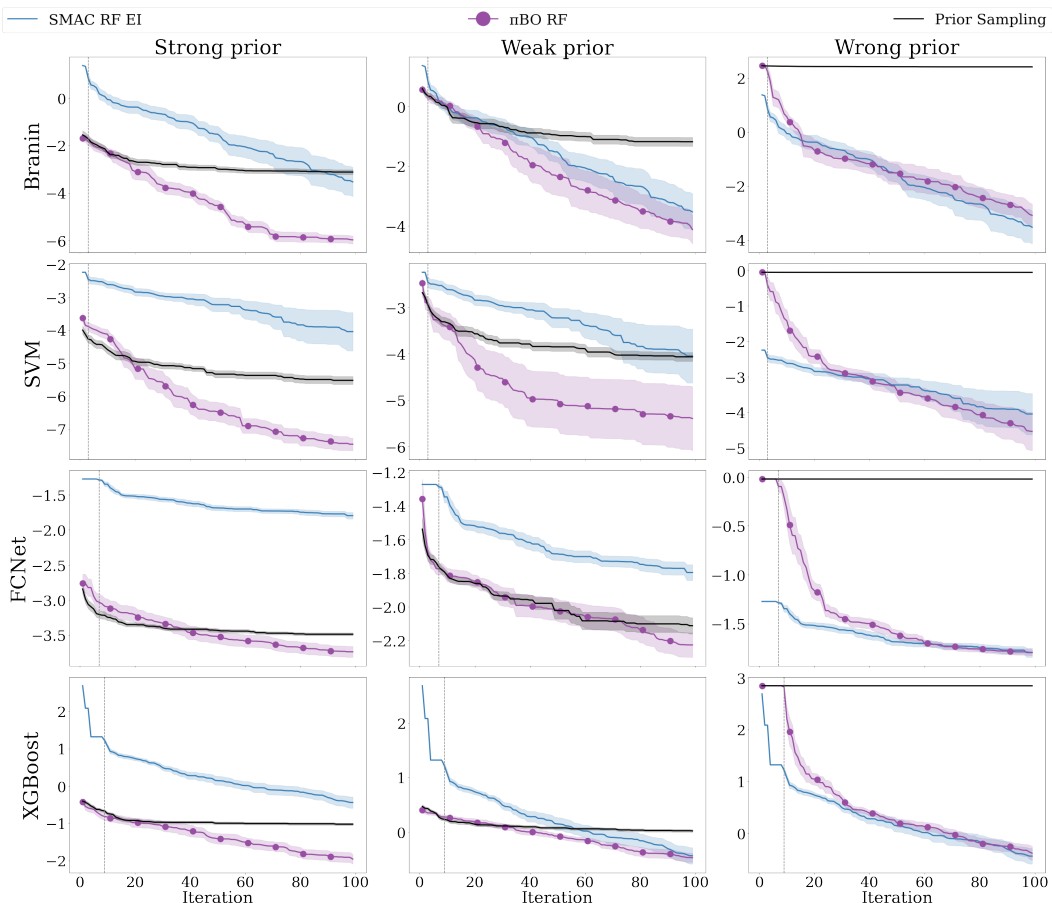

Figure 8: $\pi$BO with binning and SMAC using and RF surrogate and EI acquisition function, $\beta = 10$, and the supporting frameworks for Branin and Profet benchmarks for varying prior qualities. The mean and standard error of log simple regret is displayed over 100 iterations, averaged over 10 repetitions. Iteration 1 is removed for visibility purposes. The vertical line represents the end of the initial design phase.

## D    PRIOR CONSTRUCTION

We now present the method by which we construct our priors. For the synthetic benchmarks, we mimic (Souza et al., 2021) by offsetting a Gaussian distribution from the optima. For our case studies, we choose a Gaussian prior with zero correlation between dimensions. This was required in order to have a simple, streamlined approach that was compatible with all frameworks. We constructed the priors once before conducting the experiments, and kept them fixed throughout.

**Synthetic and Surrogate-based HPO Benchmarks**    For these benchmarks, the approximate optima of all included functions could be obtained in advance, either analytically or empirically through extensive sampling. Thus, the correctness of the prior is ultimately known in advance. For a function of dimensionality $d$ with optimum at $\boldsymbol{x^*}$, the strong and weak prior qualities were constructed by using a quality-specific noise term $\boldsymbol{\epsilon} = \{\epsilon_i\}_{i=1}^d$ and quality-specific standard deviation as a fraction of the search space. For the strong prior $\pi_s(\boldsymbol{x})$, we use a small standard deviation $\sigma_s = 1\%$ and construct the prior as

$$\pi_s(\boldsymbol{x}) \sim \mathcal{N}(\boldsymbol{x^*} + \boldsymbol{\epsilon}, \sigma_s), \quad \epsilon_i \sim \mathcal{N}(0, \sigma_s). \tag{10}$$

We construct the weak priors analogously by using a larger standard deviation $\sigma_w = 10\%$. For our 20 runs of the strong and weak prior, this procedure yielded us 20 unique priors per quality type, with varying offsets from the true optimum. Additionally, the density on the optimum is substantially

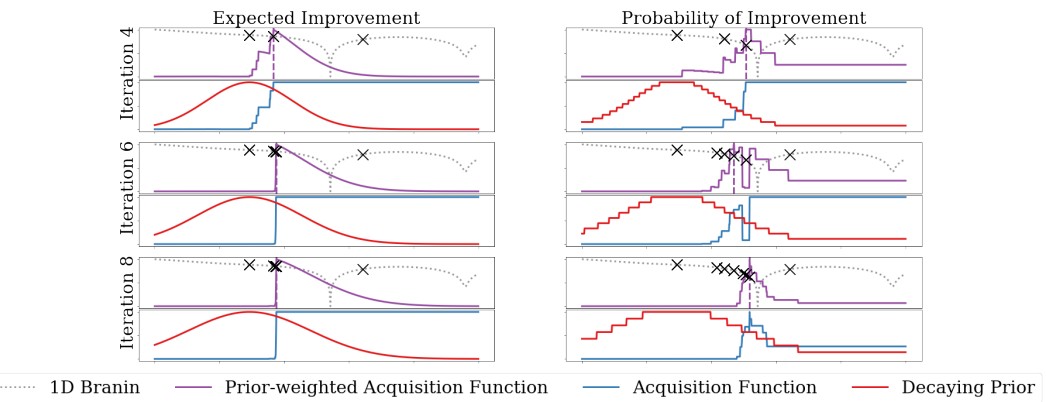

Figure 9: Point selection strategy evolution for πBO on SMAC with a RF surrogate, when employing a smooth prior (left) and binned prior (right) for the 1D-Branin function. The plots show rescaled values of prior-weighted EI (purple), EI (blue) and $\pi_n$ (red) on a 1D-Branin (grey) with global optimum in the center of the search space and the current selection as a vertical violet line. The rows represent iterations 5, 10, 15 and 20, for a run with $\beta = 5$.

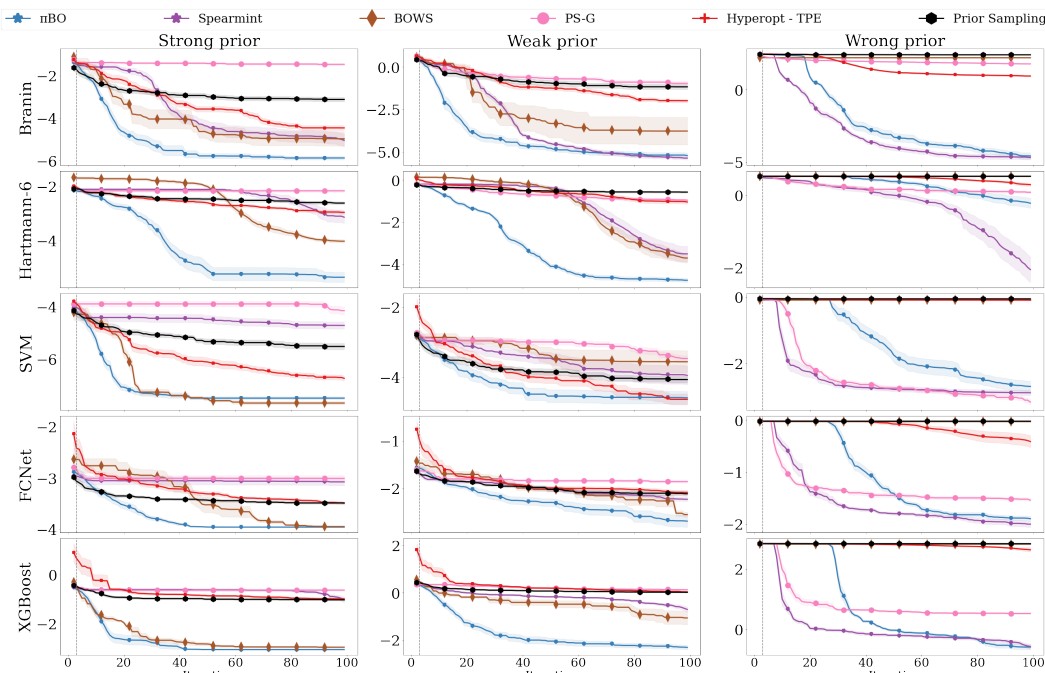

Figure 10: Comparison of πBO, Spearmint and other approaches for priors over the optimum for Branin, Hartmann and Profet benchmarks. πBO is implemented in Spearmint. The mean and standard error of log simple regret is displayed over 100 iterations, averaged over 20 repetitions. Iteration 1 is removed for visibility purposes. The vertical line represents the end of the initial design phase.

larger for the strong prior than the weak prior. No priors with a mean outside the search space were allowed, such priors were simply replaced. For Branin, we only considered one of the three Branin optima for this procedure, since not all included frameworks support multi-modal distributions. For the wrong prior, we construct it similarly to the strong prior, but around the empirical maximum, $\boldsymbol{x}^{\bar{*}}$, of the objective function in the search space. Since this point was far away from the optimum for all benchmarks, we did not add additional noise. So, the wrong prior $\pi_m$ is constructed as

$$\pi_m(\boldsymbol{x}) \sim \mathcal{N}(\boldsymbol{x}^{\bar{*}}, \sigma_s), \tag{11}$$

which means that the wrong prior is identical across runs for a given benchmark.

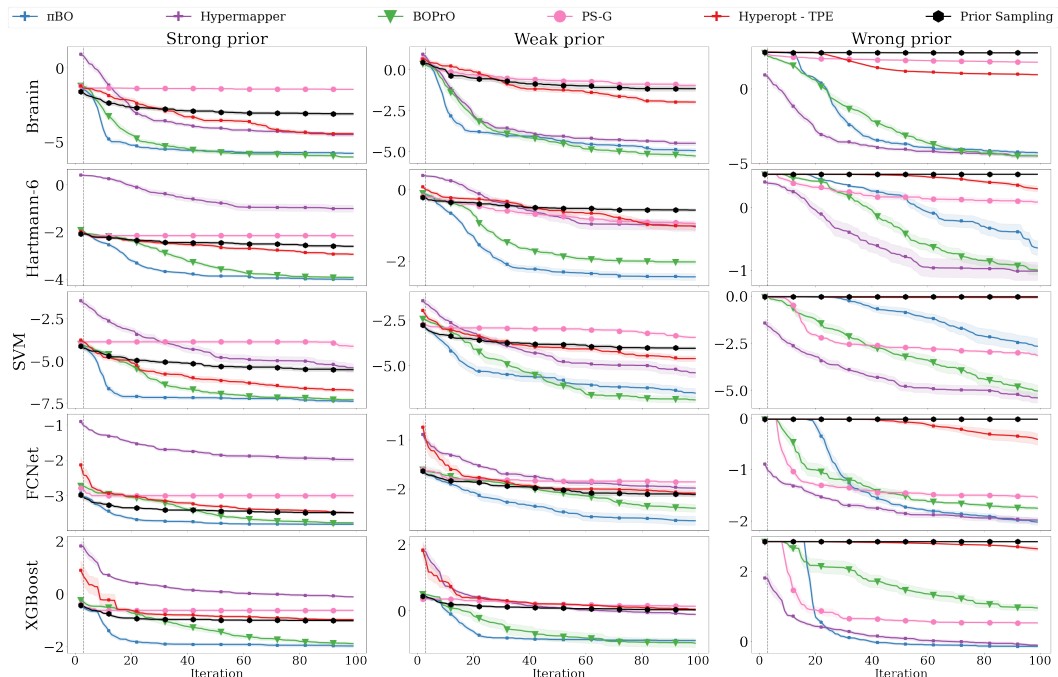

Figure 11: Comparison of $\pi$BO, Hypermapper and other approaches for priors over the optimum for Branin, Hartmann and Profet benchmarks. $\pi$BO is implemented in Spearmint. The mean and standard error of log simple regret is displayed over 100 iterations, averaged over 20 repetitions. Iteration 1 is removed for visibility purposes. The vertical line represents the end of the initial design phase.

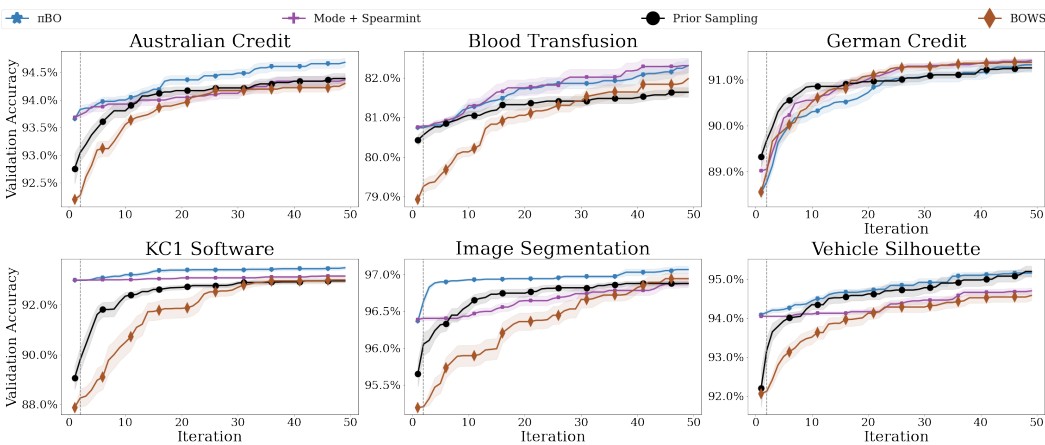

Figure 12: Comparison of $\pi$BO, BOPrO, BOWS, and prior sampling for 5D MLP tuning on various OpenML datasets for a prior centered around default values. We show mean and standard error of the accuracy across 20 repetitions. Iteration 1 is removed for visibility purposes. The vertical line represents the end of the initial design phase.

## E  PROOFS

Here, we provide the complete proofs for the Theorem and Corollary introduced in 3.3. In addition, we provide insight into the interplay between $\beta$, the prior $\pi$, and the value of the derived bound $C_{\pi,n}$.

**Theorem 1.** *Given $\mathcal{D}_n$, $K_{\boldsymbol{\ell}}$, $\pi$, $\sigma$, $\boldsymbol{\ell}$, $R$ and the compact set $\mathcal{X} \subset \mathbb{R}^d$ as defined above, the loss $\mathcal{L}_n$ incurred at iteration $n$ by $EI_{\pi,n}$ can be bounded from above as*

$$\mathcal{L}_n(EI_{\pi,n}, \mathcal{D}_n, \mathcal{H}_{\boldsymbol{\ell}}(\mathcal{X}), R) \leq C_{\pi,n} \mathcal{L}_n(EI_n, \mathcal{D}_n, \mathcal{H}_{\boldsymbol{\ell}}(\mathcal{X}), R), \quad C_{\pi,n} = \left( \frac{\max_{\boldsymbol{x} \in \mathcal{X}} \pi(\boldsymbol{x})}{\min_{\boldsymbol{x} \in \mathcal{X}} \pi(\boldsymbol{x})} \right)^{\beta/n}. \tag{12}$$

*Proof.* To bound the performance of $EI_\pi$ to that of EI, we primarily need to consider Lemma 7 and Lemma 8 by Bull Bull (2011). In Lemma 7, it is stated that for any sequence of points $\{\boldsymbol{x}_i\}_{i=1}^n$, dimensionality $d$, kernel length scales $\boldsymbol{\ell}$, and $p \in \mathbb{N}$, the posterior variance $s_n^2$ on $\boldsymbol{x}_{n+1}$ will, for a large value $C$, satisfy the following inequality at most $p$ times,

$$s_n(\boldsymbol{x}_{n+1}; \boldsymbol{\ell}) \geq Cp^{-(\nu \wedge 1)/d}(\log p)^\gamma, \quad \gamma = \begin{cases} \alpha, & \nu \leq 1 \\ 0, & \nu > 1 \end{cases}. \tag{13}$$

Thus, we can bound the posterior variance by assuming a point in time $n_p$ where Eq. 13 has held $p$ times. We now consider Lemma 8 where, through a number of inequalities, EI is bounded by the actual improvement $I_n$

$$\max \left( I_n - Rs, \frac{\tau(-R/\sigma)}{\tau(R/\sigma)} I_n \right) \leq EI_n(\boldsymbol{x}) \leq I_n + (R + \sigma)s, \tag{14}$$

where $I_n = (f(\boldsymbol{x}_n^*) - f(\boldsymbol{x}))^+$, $\tau(z) = z\Phi(z) + \phi(z)$ and $s = s_n(\boldsymbol{x}_n; \boldsymbol{\ell})$. Since $\pi$BO re-weights $EI_n$ by $\pi_n$, these bounds need adjustment to hold for $EI_{\pi,n}$. For the upper bound provided in Lemma 8, we make use of $\max_{\boldsymbol{x} \in \mathcal{X}} \pi_n(\boldsymbol{x})$ to bound $EI_{\pi,n}(\boldsymbol{x})$ for any point $\boldsymbol{x} \in \mathcal{X}$:

$$\frac{EI_{\pi,n}(\boldsymbol{x})}{\max_{\boldsymbol{x} \in \mathcal{X}} \pi_n(\boldsymbol{x})} = \frac{EI_n(\boldsymbol{x})\pi_n(\boldsymbol{x})}{\max_{\boldsymbol{x} \in \mathcal{X}} \pi_n(\boldsymbol{x})} \leq EI_n(\boldsymbol{x}) \leq I_n + (R + \sigma)s. \tag{15}$$

For the lower bounds, we instead rely on $\min_{\boldsymbol{x} \in \mathcal{X}} \pi_n(\boldsymbol{x})$ in a similar manner:

$$\max \left( I_n - Rs, \frac{\tau(-R/\sigma)}{\tau(R/\sigma)} I_n \right) \leq EI_n(\boldsymbol{x}) \leq \frac{EI_n(\boldsymbol{x})\pi_n(\boldsymbol{x})}{\min_{\boldsymbol{x} \in \mathcal{X}} \pi_n(\boldsymbol{x})} = \frac{EI_{\pi,n}(\boldsymbol{x})}{\min_{\boldsymbol{x} \in \mathcal{X}} \pi_n(\boldsymbol{x})}. \tag{16}$$

Consequently, $EI_\pi$ can be bounded by the actual improvement as

$$\min_{\boldsymbol{x} \in \mathcal{X}} \pi_n(\boldsymbol{x}) \max \left( I_n - Rs, \frac{\tau(-R/\sigma)}{\tau(R/\sigma)} I_n \right) \leq EI_{\pi,n}(\boldsymbol{x}) \leq \max_{\boldsymbol{x} \in \mathcal{X}} \pi_n(\boldsymbol{x})(I_n + (R + \sigma)s). \tag{17}$$

With these bounds in place, we consider the setting as in the proof for Theorem 2 in Bull Bull (2011), which proves an upper bound for the EI strategy in the fixed kernel parameters setting. At an iteration $n_p, p \leq n_p \leq 3p$, the posterior variance will be bounded by $Cp^{-(\nu \wedge 1)/d}(\log p)^\gamma$. Furthermore, since $I_n \geq 0$ and $||f||_{\mathcal{H}_{\boldsymbol{\ell}}(\mathcal{X}))} \leq R$, we can bound the total improvement as

$$\sum_i I_i \leq \sum_i f(\boldsymbol{x}_i^*) - f(\boldsymbol{x}_{i+1}^*) \leq f(\boldsymbol{x}_1^*) - \min f \leq 2||f||_\infty \leq 2R, \tag{18}$$

leaving us a maximum of $p$ times that $I_n \geq 2Rp^{-1}$. Consequently, both the posterior variance $s_{n_p}^2$ and the improvement $I_{n_p}$ are bounded at $n_p$. For a future iteration $n$, $3p \leq n \leq 3(p+1)$, we use the bounds on $EI_\pi$, $s_{n_p}$ and $I_{n_p}$ to obtain the bounds on the $EI_\pi$ loss:

$$\mathcal{L}_n(EI_\pi, \mathcal{D}_n, \mathcal{H}_{\boldsymbol{\ell}}(\mathcal{X}), R)$$
$$= f(\boldsymbol{x}_n^*) - \min f$$
$$\leq f(\boldsymbol{x}_{n_p}^*) - \min f$$
$$\leq \frac{EI_{\pi,n_p}(\boldsymbol{x}^*)}{\min_{\boldsymbol{x} \in \mathcal{X}} \pi_n(\boldsymbol{x})} \frac{\tau(R/\sigma)}{\tau(-R/\sigma)}$$
$$\leq \frac{EI_{\pi,n_p}(\boldsymbol{x}_{n+1})}{\min_{\boldsymbol{x} \in \mathcal{X}} \pi_n(\boldsymbol{x})} \frac{\tau(R/\sigma)}{\tau(-R/\sigma)}$$
$$\leq \frac{\max_{\boldsymbol{x} \in \mathcal{X}} \pi_n(\boldsymbol{x})}{\min_{\boldsymbol{x} \in \mathcal{X}} \pi_n(\boldsymbol{x})} \frac{\tau(R/\sigma)}{\tau(-R/\sigma)} \left( I_{n_p} + (R + \sigma)s_{n_p} \right)$$
$$\leq \left( \frac{\max_{\boldsymbol{x} \in \mathcal{X}} \pi(\boldsymbol{x})}{\min_{\boldsymbol{x} \in \mathcal{X}} \pi(\boldsymbol{x})} \right)^{\beta/n} \frac{\tau(R/\sigma)}{\tau(-R/\sigma)} (2Rp^{-1} + (R + \sigma)Cp^{-(\nu \wedge 1)/d}(\log p)^\gamma),$$

where the last inequality is a factor $C_{\pi,n} = \left(\frac{\max_{\boldsymbol{x}\in\mathcal{X}}\pi(\boldsymbol{x})}{\min_{\boldsymbol{x}\in\mathcal{X}}\pi(\boldsymbol{x})}\right)^{\beta/n}$ larger than the bound on $\mathcal{L}_n(\text{EI},\mathcal{D}_n,\mathcal{H}_{\boldsymbol{\ell}}(\mathcal{X}),R)$. $\qquad\square$

**Corollary 1.** *The loss of a decaying prior-weighted Expected Improvement strategy, $EI_\pi$, is asymptotically equal to the loss of an Expected Improvement strategy, EI:*

$$\mathcal{L}_n(EI_{\pi,n},\mathcal{D}_n,\mathcal{H}_{\boldsymbol{\ell}}(\mathcal{X}),R) \sim \mathcal{L}_n(EI_n,\mathcal{D}_n,\mathcal{H}_{\boldsymbol{\ell}}(\mathcal{X}),R), \tag{19}$$

*so we obtain a convergence rate for $EI_\pi$ of $\mathcal{L}_n(EI_{\pi,n},\mathcal{D}_n,\mathcal{H}_{\boldsymbol{\ell}}(\mathcal{X}),R) = \mathcal{O}(n^{-(\nu\wedge 1)/d}(\log n)^\gamma)$.*

*Proof.* We simply compute the fraction of the losses in the limit,

$$\lim_{n\to\infty}\frac{\mathcal{L}_n(\text{EI}_\pi,\mathcal{D}_n,\mathcal{H}_{\boldsymbol{\ell}}(\mathcal{X}),R)}{\mathcal{L}_n(\text{EI},\mathcal{D}_n,\mathcal{H}_{\boldsymbol{\ell}}(\mathcal{X}),R)} \leq \lim_{n\to\infty}\left(\frac{\max_{\boldsymbol{x}\in\mathcal{X}}\pi(\boldsymbol{x})}{\min_{\boldsymbol{x}\in\mathcal{X}}\pi(\boldsymbol{x})}\right)^{\beta/n} = 1. \tag{20}$$

$\qquad\square$

### E.1 SENSITIVITY ANALYSIS ON $C_{\pi,n}$

We now provide additional insight into how $C_{\pi,n}$ depends on the choices of prior and $\beta$ made by the user. To do so, we consider a typical low-budget setting and display values of $C_{\pi,n}$ at iteration 50. We consider a one-dimensional search space where with a Gaussian prior located in the center of the search space. In the plot below, we display how the choice of $\sigma$, given as a percentage of the search space, and $\beta$, the prior confidence parameter, yield different values of $C_{\pi,n}$. We see that, for approximately half of the space, the upper bound on the loss is at least 80% (bright green or yellow) of the upper bound of EI, and only a small region of very narrow priors (dark blue) give a low guaranteed convergence rate.

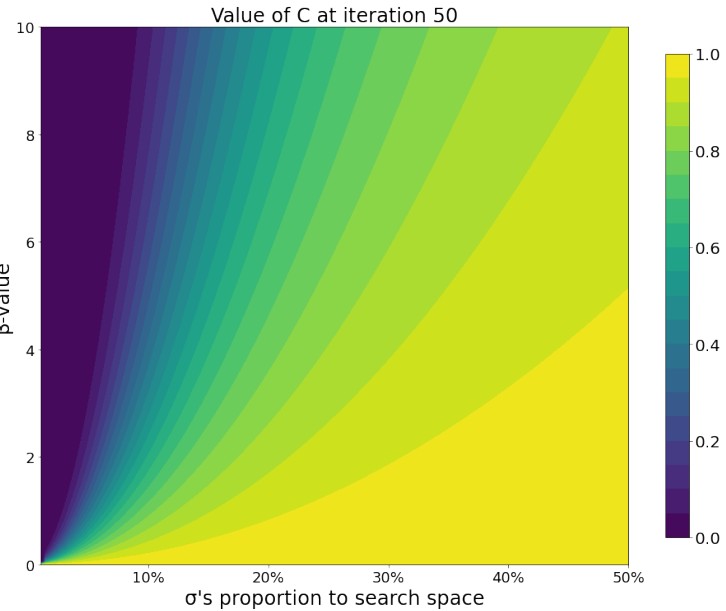

Figure 13: Value of $C_{\pi,n}$ at iteration 50 for a centered Gaussian prior in a one-dimensional search space, varying $\sigma$ and $\beta$. The value of $C_{\pi,n}$ upper bounds the loss incurred by $EI_\pi$ relative EI at the at the given iteration, given the same data. Regions in dark represent pairs of $(\sigma,\beta)$ where the upper bound on prior-weighted EI is low relative to EI, whereas regions in yellow represent pairs of $(\sigma,\beta)$ where the upper bound is approximately the same.

# F    EXPERIMENT DETAILS

## F.1    FRAMEWORKS

Our implementations of $\pi$BO require little change in the supporting frameworks, Spearmint and HyperMapper, and we stay as close to the default settings as possible for each framework. For both Spearmint and HyperMapper, we consider a Matérn 5/2 Kernel. For particularly strong priors, rounding errors can cause the prior to be zero in parts of the search space, potentially affecting $\pi$BO's convergence properties. To avoid these rounding errors and ensure a strictly positive prior, we add a small constant, $\epsilon = 10^{-12}$, to the prior throughout the search space for all prior qualities. For the initial sampling from the prior, we truncate the distribution by disallowing sampled points from outside the search space, instead re-sampling such points. During optimization, we do not to explicitly truncate the prior, as points outside the search space are never considered during acquisition function maximization. Thus, the prior is effectively truncated to fit the search space without requiring additional consideration.

To the best of our knowledge, there is no publicly available implementation of BOWS, so we re-implemented it in Spearmint. For the Spearmint implementation of BOWS, we provide warped versions of each benchmark, obtaining 20 unique warpings per prior quality and benchmark. We truncate the prior by restricting the warped search space to only include the region which maps back to the original search space through the inverted warping function. For all other approaches, we use the original, publicly available implementations. Notably, the available implementation of Hyperopt TPE does not support bounded search spaces under our priors; as a compromise, when asked to evaluate outside the search space we return an empirically obtained maximum on the objective function inside the search space.

We use the search spaces, prior locations and descriptions used by (Souza et al., 2021) for the toy and surrogate HPO problems. We now provide additional details about the benchmarks and case study tasks used, their associated search spaces and priors, and the resources used to run these studies.

## F.2    BENCHMARKS AND CASE STUDIES

**Branin**    The Branin function is a well-known synthetic benchmark for optimization problems. The Branin function has two input dimensions and three global minima.

**Hartmann-6**    The Hartmann-6 function is a well-known synthetic benchmark for optimization problems, which has one global optimum and six dimensions.

**SVM**    A hyperparameter-optimization benchmark in 2D based on Profet (Klein et al., 2019). This benchmark is generated by a generative meta-model built using a set of SVM classification models trained on 16 OpenML tasks. The benchmark has two input parameters, corresponding to SVM hyperparameters.

**FCNet**    A hyperparameter and architecture optimization benchmark in 6D based on Profet. The FC-Net benchmark is generated by a generative meta-model built using a set of feed-forward neural networks trained on the same 16 OpenML tasks as the SVM benchmark. The benchmark has six input parameters corresponding to network hyperparameters.

**XGBoost**    A hyperparameter-optimization benchmark in 8D based on Profet. The XGBoost benchmark is generated by a generative meta-model built using a set of XGBoost regression models in 11 UCI datasets. The benchmark has eight input parameters, corresponding to XGBoost hyperparameters.

**OpenML MLP**    The OpenML MLP tuning tasks are provided through HPOBenchEggensperger et al. (2021), and train binary classifiers on real-world datasets. The 5D parameter space consists of four continous parameters and one integer parameter.

**U-Net Medical**    The U-Net (Ronneberger et al., 2015) is a popular convolutional neural network architecture for image segmentation. We use the implementation and evaluation setting from the popular NVIDIA deep learning examples repository (Przemek et al.) to build a case study for optimizing hyperparameters for U-Net. The NVIDIA repository is aimed towards the segmentation of neuronal processes in electron microscopy images for the 2D EM segmentation challenge dataset

Table 1: Search spaces for the Branin and Profet benchmarks. We report the original ranges and whether or not a log scale was used. Mean of the strong and weak priors before offset are reported.

| Benchmark | Range | Parameter values | Log scale | Prior mean |
|---|---|---|---|---|
| Branin | $x_1$ | $[-5, 10]$ | - | $\pi$ |
| | $x_2$ | $[0, 15]$ | - | 2.275 |
| SVM | C | $[e^{-10}, e^{10}]$ | ✓ | $e^{7.84}$ |
| | $\gamma$ | $[e^{-10}, e^{10}]$ | ✓ | $e^{-9.35}$ |
| FCNet | learning rate | $[10^{-6}, 10^{-1}]$ | ✓ | $10^{-6}$ |
| | batch size | $[8, 128]$ | ✓ | 8.7 |
| | units layer 1 | $[16, 512]$ | ✓ | 210 |
| | units layer 2 | $[16, 512]$ | ✓ | 205 |
| | dropout rate l1 | $[0.0, 0.99]$ | - | 0.0007 |
| | dropout rate l2 | $[0.0, 0.99]$ | - | 0.852 |
| XGBoost | learning rate | $[10^{-6}, 10^{-1}]$ | ✓ | $10^{-6}$ |
| | gamma | $[0, 2]$ | - | 2 |
| | L1 regularization | $[10^{-5}, 10^3]$ | ✓ | $10^{0.088}$ |
| | L2 regularization | $[10^{-5}, 10^3]$ | ✓ | $10^{-1.78}$ |
| | number of estimators | $[10, 500]$ | - | 500 |
| | subsampling | $[0.1, 1]$ | - | 0.1 |
| | maximum depth | $[1, 15]$ | - | 1 |
| | minimum child weight | $[0, 20]$ | - | 8.42 |

Table 2: Search spaces for the two case studies. We report the original ranges and whether or not a log scale was used. Mean and standard deviation of the priors are reported, where the standard deviation is reported as a percentage of the search space. For the categorical variable pooling, the probabilities for activation functions in U-Net were set to uniform and the probabilities for pooling, [Avg, Max], were set to $[0.2, 0.8]$, respectively.

| Benchmark | Parameter name | Range | Log scale | Prior mean | Prior st.dev. |
|---|---|---|---|---|---|
| OpenML MLP | alpha | $[10^{-6}, 10^{-2}]$ | ✓ | $10^{-4}$ | 25% |
| | batch size | $[10^{-6}, 10^{-2}]$ | ✓ | $10^{-4.5}$ | 25% |
| | depth | $\{1, 2, 3\}$ | - | 0.5 | 25% |
| | initial learning rate | $[0.6, 0.99]$ | ✓ | 0.9 | 25% |
| | width | $[0.9, 0.9999]$ | ✓ | 0.999 | 25% |

(Arganda-Carreras et al., 2015; Cardona et al., 2010). We optimize 6 hyperparameters of the U-Net pipeline.

**ImageNette** ImageNette (Howard, 2019) is a subset of 10 classes of ImageNet (Deng et al., 2009) and is primarily used for algorithm development for the popular FastAI library (Howard et al., 2018). The FastAI library contains a convolutional neural network pipeline for ImageNette, that is used by all competitors on the ImageNette leaderboard. We base our case study on the 80 epoch, 128 resolution setting of this leaderboard and optimize 6 of the hyperparameters of the FastAI ImageNette pipeline.

## F.3 SEARCH SPACES AND PRIORS

The search spaces for each benchmark are summarized in Table 1 (Branin and Profet), Table 2 (OpenML MLP), and Table 3 (ImageNette and U-Net). For the Profet benchmarks, we report the original ranges and whether or not a log scale was used. However, in practice, Profet's generative model transforms the range of all hyperparameters to a linear $[0, 1]$ range. We use Emukit's public implementation for these benchmarks (Paleyes et al., 2019).

Table 3: Search spaces for the two case studies. We report the original ranges and whether or not a log scale was used. Mean and standard deviation of the priors are reported, where the standard deviation is reported as a percentage of the search space. For the categorical variable pooling, the probabilities for activation functions in U-Net were set to uniform and the probabilities for pooling, [Avg, Max], were set to [0.2, 0.8], respectively.

| Benchmark | Parameter name | Range | Log scale | Prior mean | Prior st.dev. |
|---|---|---|---|---|---|
| U-Net Medical | learning rate | $[10^{-6}, 10^{-2}]$ | ✓ | $10^{-4}$ | 25% |
| | weight decay | $[10^{-6}, 10^{-2}]$ | ✓ | $10^{-4.5}$ | 25% |
| | dropout | $[0, 0.99]$ | - | 0.5 | 25% |
| | $\beta_1$ | $[0.6, 0.99]$ | ✓ | 0.9 | 25% |
| | $\beta_2$ | $[0.9, 0.9999]$ | ✓ | 0.999 | 25% |
| | activation | 3 options[2] | - | - | 25% |
| ImageNette-128 | learning rate | $[10^{-4}, 0]$ | ✓ | $10^{-2.10}$ | 25% |
| | squared momentum | $[0.9, 0.999]$ | ✓ | 0.99 | 25% |
| | momentum | $[0, 0.99]$ | - | 0.95 | 25% |
| | epsilon | $[10^{-7}, 10^{-5}]$ | ✓ | $10^{-6}$ | 25% |
| | mixup | $[0.0, 0.5]$ | - | 0.4 | 25% |
| | pooling | {Avg, Max} | - | - | - |

## F.4 CASE STUDY DETAILS

**Training details deep learning case studies**   Both case studies are based on existing deep learning code, whose hyperparameters we vary according to the HPO. In both case studies, we enabled mixed precision training, and for ImageNette-128 to work in conjunction with Spearmint, we had to enable the MKL_SERVICE_FORCE_INTEL environment flag. For all further details, we refer to the supplementary material containing our code.

**Resources used for deep learning case studies**   For U-Net Medical we used one GeForce RTX 2080 Ti GPU, whereas for ImageNette-128 we used two GeForce RTX 2080 Ti GPU's. Also, we used 4 cores and 8 cores respectively, of an AMD EPYC 7502 32-Core Processor. In Table 4 we list the GPU hours needed for running the deep learning case studies as well as the emitted $CO_2$ equivalents.

Table 4: Approximate GPU hours required to perform an evaluation for one approach for the deep learning case studies. Additionally we report approximate carbon footprints using the MachineLearning Impact calculator presented in Lacoste et al. (2019) based on OECD's 2014 yearly average carbon efficiency.

| Benchmark | Repetitions | GPUh's Per Repetition | Total GPUh | Total kgCO$_2$eq |
|---|---|---|---|---|
| U-Net Medical | 20 | 9 | 180 | 19.4 |
| ImageNette-128 | 10 | 40 | 400 | 43.2 |

**Assets deep learning case studies**   In addition to the assets we list in the main paper, the U-Net Medical code base we used employs the 2D EM segmentation challenge dataset (Arganda-Carreras et al., 2015; Cardona et al., 2010), which is available for for the purpose of generating or testing non-commercial image segmentation software. We include licenses of all existing code assets we used in the supplementary material containing our code.

## G   SENSITIVITY TO PRIOR STRENGTH

We investigate the performance of $\pi$BO when providing priors over the optimum of various qualities. To show the effect of decreasing the prior strength, a grid of prior qualities, with varying widths and

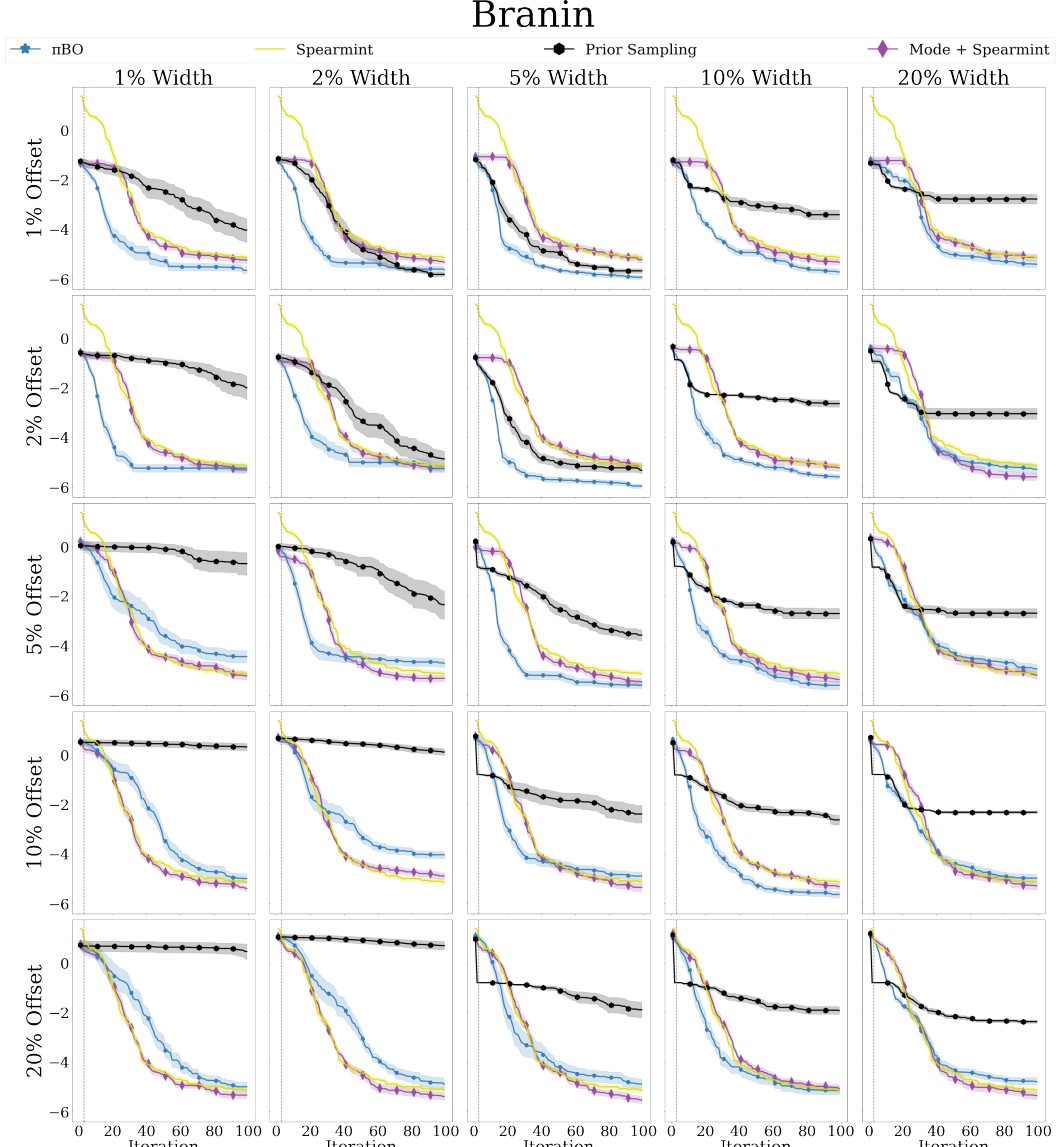

Figure 14: Comparison on Branin of priors with varying widths and offsets over the optimum on $\pi$BO, Spearmint, Spearmint with mode initialization, and sampling from the prior. The mean and standard error of log simple regret is displayed over 100 iterations, averaged over 20 repetitions. Iteration 1 is removed for visibility purposes.

offsets from the optimum, are provided. Thus, priors range from the strong prior in the results, to weak, correct priors and sharp, misplaced priors.

From Figures 14- 18, it is shown that $\pi$BO provides substantial performance across most prior qualities for all benchmarks but Branin, and recoups its early losses on the worst priors in the bottom left corner. $\pi$BO demonstrates sensitivity to the width of the prior, as the optimization does not progress as quickly for well-located priors with a larger width. Additionally, $\pi$BO's improvement over the Spearmint + Mode baseline is further emphasized, as this baseline often fails to meaningfully improve over the mode in early iterations.

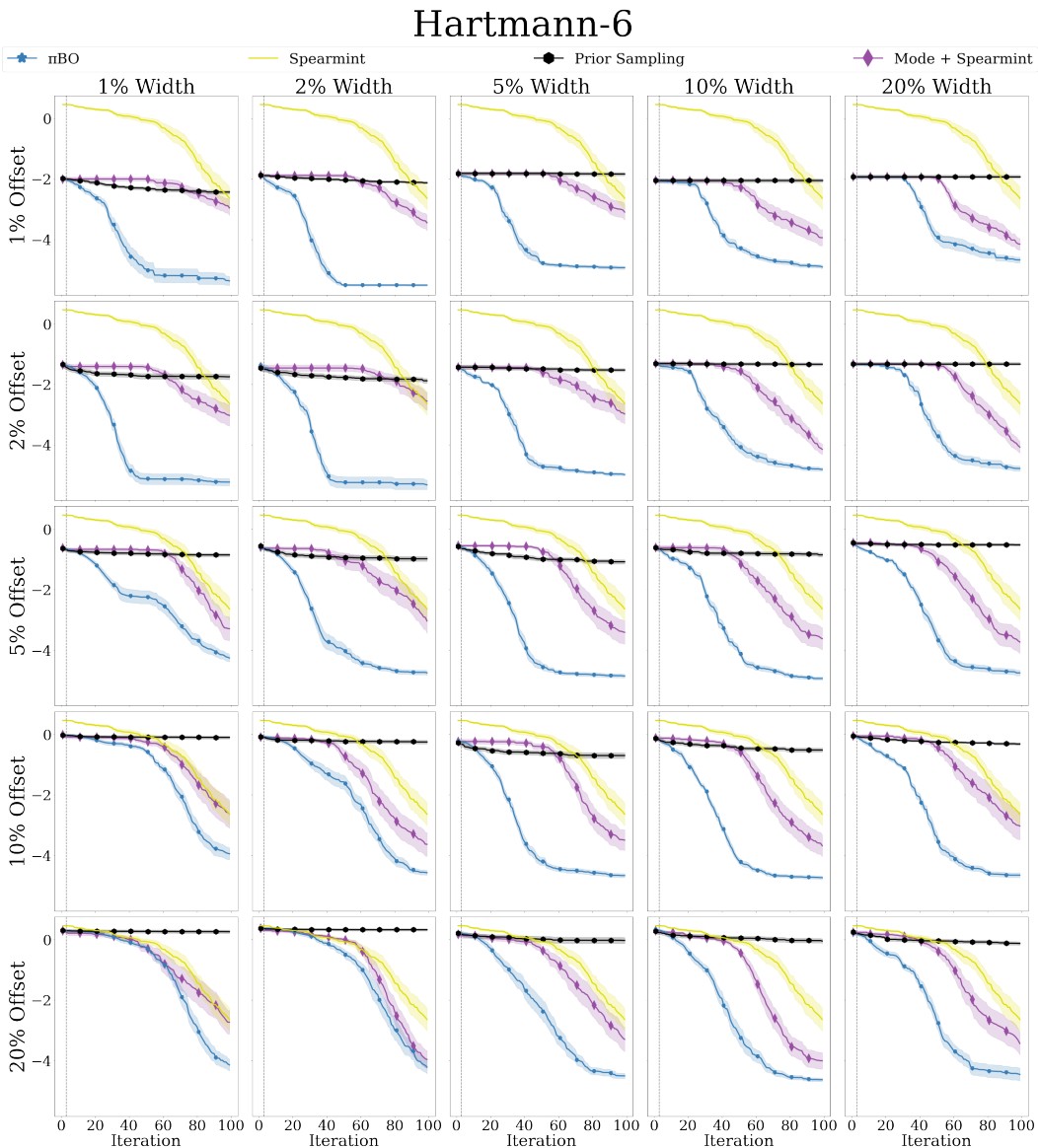

Figure 15: Comparison on Hartmann-6 of priors with varying widths and offsets over the optimum on πBO, Spearmint, Spearmint with mode initialization, and sampling from the prior. The mean and standard error of log simple regret is displayed over 100 iterations, averaged over 20 repetitions. Iteration 1 is removed for visibility purposes.

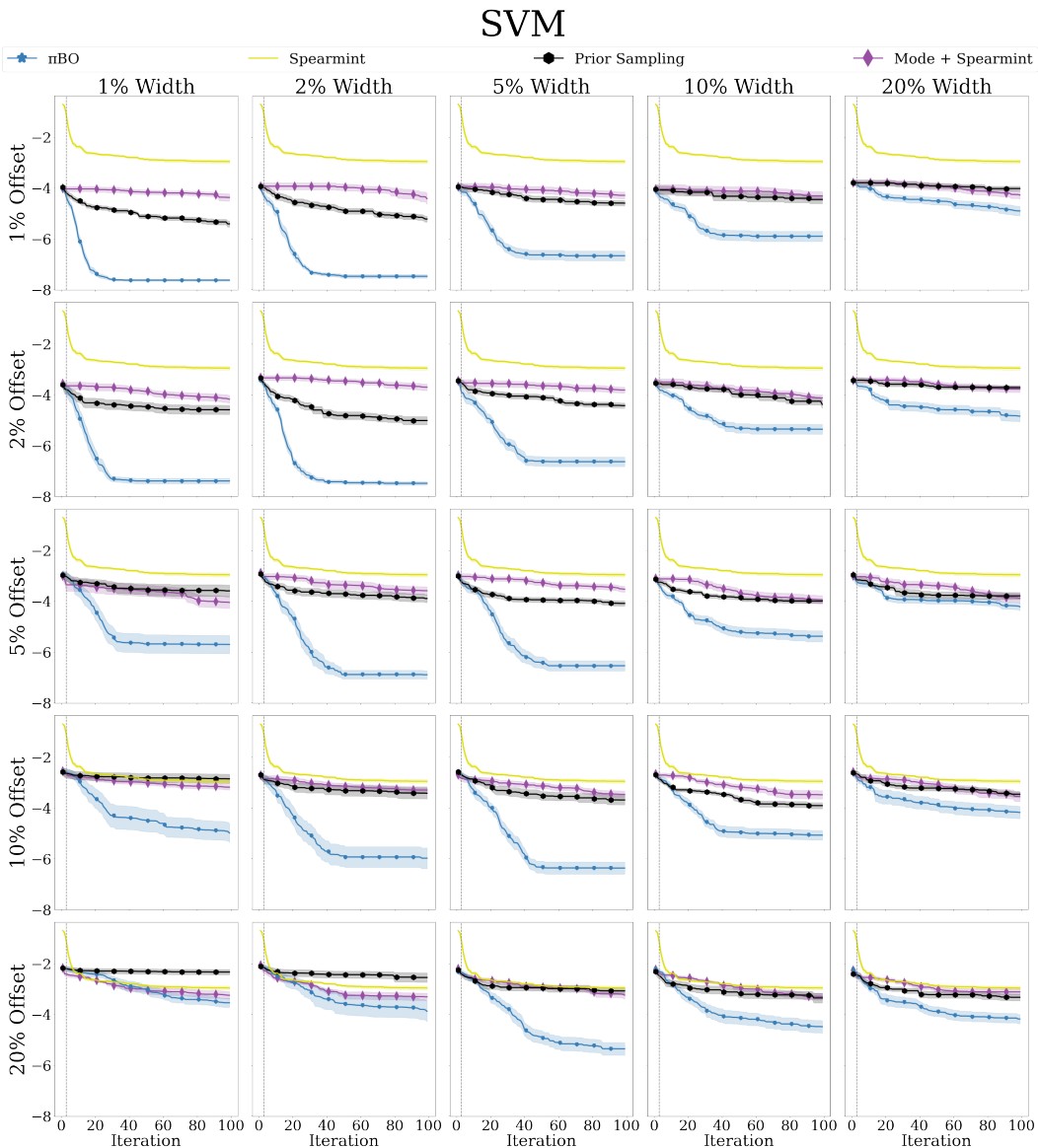

Figure 16: Comparison on SVM of priors with varying widths and offsets over the optimum on $\pi$BO, Spearmint, Spearmint with mode initialization, and sampling from the prior. The mean and standard error of log simple regret is displayed over 100 iterations, averaged over 20 repetitions. Iteration 1 is removed for visibility purposes.

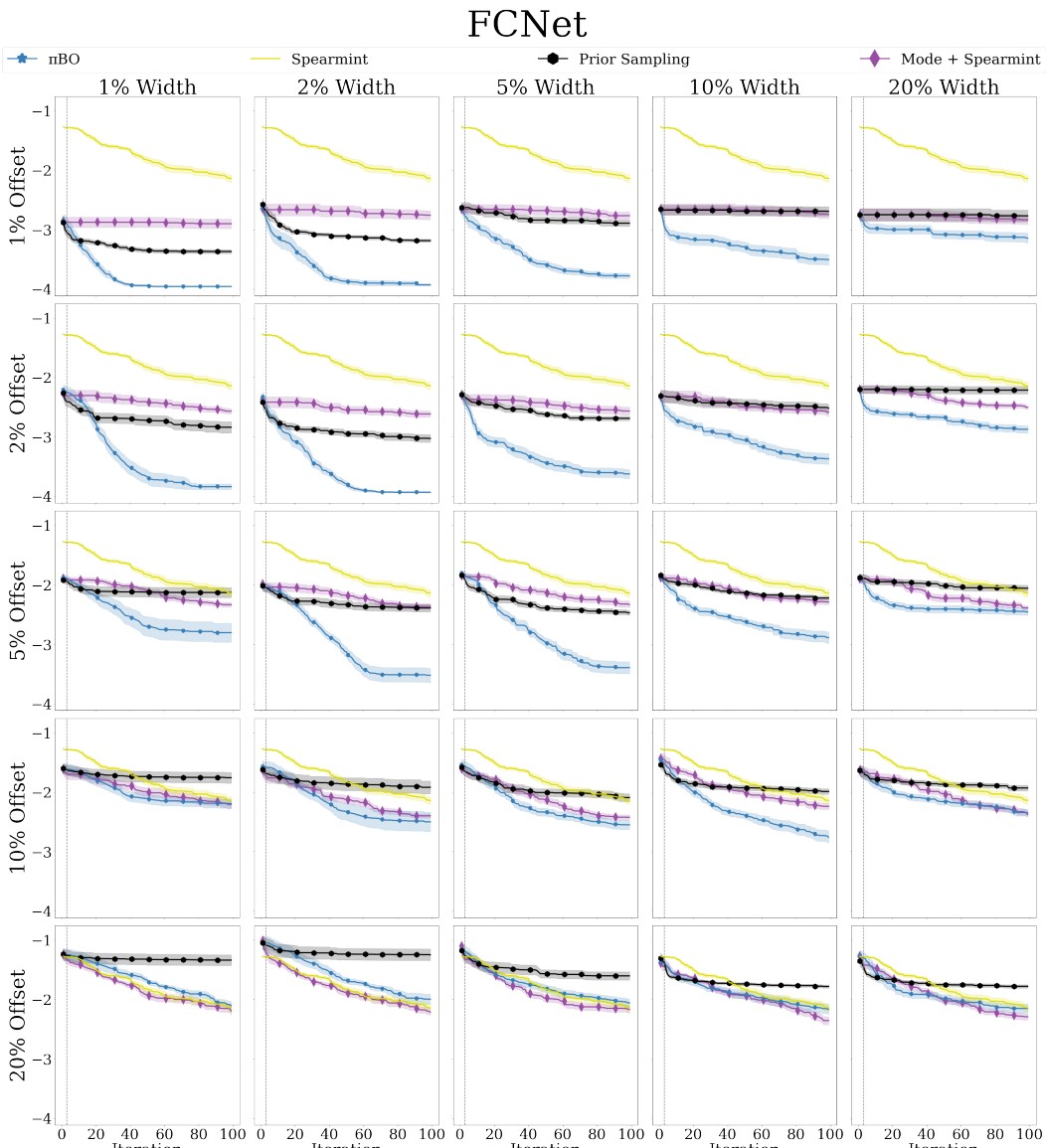

Figure 17: Comparison on FCNet of priors with varying widths and offsets over the optimum on πBO, Spearmint, Spearmint with mode initialization, and sampling from the prior. The mean and standard error of log simple regret is displayed over 100 iterations, averaged over 20 repetitions. Iteration 1 is removed for visibility purposes.

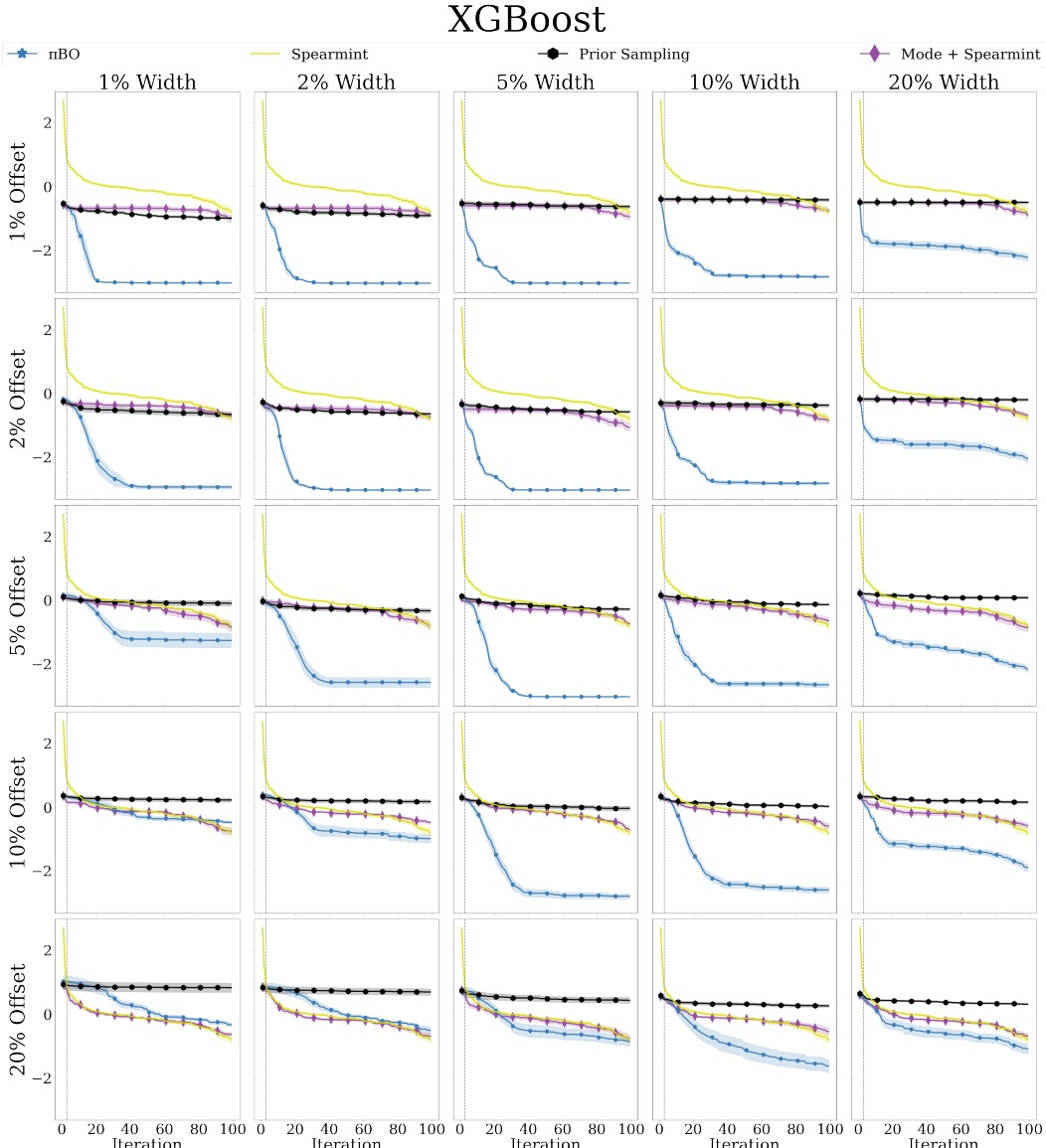

Figure 18: Comparison on XGBoost of priors with varying widths and offsets over the optimum on πBO, Spearmint, Spearmint with mode initialization, and sampling from the prior. The mean and standard error of log simple regret is displayed over 100 iterations, averaged over 20 repetitions. Iteration 1 is removed for visibility purposes.

