# OpenReview forum: "$\pi$BO: Augmenting Acquisition Functions with User Beliefs for Bayesian Optimization"
_ICLR.cc/2022/Conference — ICLR 2022 Poster_

### Official Review · Reviewer_vMEc · 2021-10-27

**Correctness:** 4
**Technical Novelty And Significance:** 2
**Empirical Novelty And Significance:** 3
**Recommendation:** 8
**Confidence:** 5

**Main Review:**

The paper is clear and with extensive empirical evaluation. An asymptotic convergence results shows that there is only a constant factor difference with the regular method. The main drawback lies in the apparent simplicity of the approach, in terms of originality. Giving more weights to areas supposed to be better in the acquisition function optimization could already exist in application oriented papers (though I did not found such example with a quick search).

Minor:
- Consistent colors/symbols for methods across figures would help.
- Additional details on how to define the prior would make the paper more self contained.
- Note that the unbounded BO method would apply with the definition of starting bounds (say taking 50% of the volume under the prior).
- The differences with BOPrO could be better highlighted.
- I appreciate the theoretical result albeit one could hope to show some improvement from using a good prior.

Typos:
P1: documentesd

**Summary Of The Paper:**

In this article the authors propose to incorporate domain knowledge on where good configurations are located in Bayesian optimization (BO). They do so by weighting the acquisition function by a prior that decays and reverts to uniform as iterations go. The asymptotic rate of convergence is shown to be of the same order as the non-prior version. Empirical tests are provided on various synthetic and  hyperparameter tuning tasks, with diverse baseline methods and ablation studies.

**Summary Of The Review:**

If not particularly original, the proposition is well motivated and linked to the state of the art, plus it comes with extensive empirical results supporting the approach.

---

> ### Author Response · Authors · 2021-11-12
> **Response to Reviewer vMEc**
>
>
>
> We are glad that the reviewer appreciates our work and our extensive empirical analysis. We also thank the reviewer for the thorough feedback, and have addressed your minor remarks.
>
> $~$
> >##### The main drawback lies in the apparent simplicity of the approach, in terms of originality.
>
> While we agree that $\pi$BO is conceptually simple, we believe that simplicity is a virtue when combined with strong empirical results. In fact, $\pi$BO’s simplicity allows it to be easily implemented in different BO frameworks and therefore, we hope it will be impactful. Furthermore, $\pi$BO has much better theoretical properties and better empirical results than all previous approaches for taking into account user beliefs over the location of the optimum.
>
> $~$
> >##### Giving more weights to areas supposed to be better in the acquisition function optimization could already exist in application oriented papers (though I did not found such example with a quick search).
>
> We agree with Reviewer eisp’s comment that it is surprising that this has not been done before, and we take this sense of surprise as encouragement.
>
> While we can not completely rule out that some application-oriented paper has presented similar work before, we believe that properly studying this method from a theoretical and comprehensive empirical point of view holds substantial merit. Moreover, the decaying of the weights is an integral part of our work, and adds novelty to the approach.
>
> $~$
> >##### Additional details on how to define the prior would make the paper more self contained.
>
> We believe that the method we employed for prior construction in Sections 4.3 and 4.4 serves as an example of how a prior can be constructed by relatively simple means. The exact details of this procedure can be found in appendix F.3. With said approach, the practitioner does not need to know an entire probability distribution, but rather one or a few points which are believed to be good. Furthermore, there are a number of ways we see a prior be formed:
>
>
>
> 1. If the prior knowledge is in the form of a small number of good solutions, a natural choice of prior could be a Gaussian KDE fitting those solutions.
> 2. If the prior knowledge is in form of “community knowledge”, such as “good values for Adam’s learning rate tend to be between $10^{-1}$ and $10^{-3}$”, then a prior covering this range is likely a good choice (e.g., a $\text{logNormal}$ prior centered at $10^{-2}$ / equivalently a Gaussian centered at $-2$ in $\text{log}_{10}$ space).
> 3. If the prior knowledge is that far-apart regions A and B might contain the optimum, but that it is unlikely to lie in between, then a multimodal prior distribution would be best (e.g., a mixture of Gaussians).
> 4. For categorical choices (e.g., choice of Adam vs. SGD vs. Nadam), a simple categorical distribution can typically be constructed by hand (the uniform distribution being equivalent to not wanting to specify a preference).
>
> In our case the prior was merely given by a default configuration found in the DL packages. This is indeed close to both (1) and (2).We also used (4) in the same experiments.
>
> $~$
> >##### I appreciate the theoretical result albeit one could hope to show some improvement from using a good prior.
>
> Although we agree that this would be very valuable, we do not currently see any way to provide reasonable bounds depending on the prior’s quality, except the following bound: if the prior’s mode is at the true optimum of the unknown function, $\pi$BO will find the optimum in the first function evaluation.
>
> The example above may seem overly trivial, but it hopefully sheds *some* light on the difficulty of defining the setting where such a prior quality-dependent bound would be constructed.
>
> **Questions:**
> >##### Note that the unbounded BO method would apply with the definition of starting bounds (say taking 50% of the volume under the prior).
>
> We were not able to discern what the reviewer meant here. Would the reviewer please enlighten us on this?
>
>
> $~$
> We thank you again for your thorough review. If the concerns were addressed, we would appreciate it if you would consider increasing your score, otherwise please do not hesitate to post additional follow-up questions.

---

> > ### Comment · Reviewer_vMEc · 2021-11-16
> > **Clarification on the unbounded question**
> >
> > Thank you for the detailed response to my comments.
> >
> > About the use of BO methods that apply for unbounded domains, they can also be used with bounded domains but starting from a region of interest (equivalent to a uniform prior in parts of the domain). Depending on the performance and application, the method may choose to extend the initial domain (potentially until the maximum domain size is reached). It does not have to be tested but it is an option.

---

> > > ### Author Response · Authors · 2021-11-17
> > > **Response to comment on unbounded BO**
> > >
> > > Thanks for the clarification.
> > >
> > > We agree with the reviewer that there is a connection between unbounded BO and our approach. We also agree that unbounded BO could potentially be translated to our problem setting and yield satisfactory results. However, we see three shortcomings of such an approach:
> > > - A translation from a prior over the optimum to a scheduling procedure for search space expansion is non-trivial.
> > > - This approach would not (at least not intrinsically) support multimodal priors.
> > > - This approach would not (at least not intrinsically)  support categorical variables.
> > >
> > > Nevertheless, we do agree that, especially for the case of numerical hyperparameters with unimodal priors, one might be able to come up with a useful approach that also could potentially enjoy theoretical guarantees. We will add a note to that effect to our paper. We thank the reviewer for this feedback. Please do not hesitate to post additional follow-up comments or questions.

---

> > > > ### Comment · Reviewer_vMEc · 2021-11-23
> > > > **Update of score**
> > > >
> > > > Based on the integration of comments and detailed response, I increased the score.

---

### Official Review · Reviewer_eisp · 2021-10-27

**Correctness:** 4
**Technical Novelty And Significance:** 2
**Empirical Novelty And Significance:** 3
**Recommendation:** 8
**Confidence:** 4

**Main Review:**

## Review:

### Strengths:
The method is simple and sensible, and experiments convincingly demonstrate that it works with a well-specified prior, and more importantly, demonstrate that a poor prior could lead to problems. I know it doesn’t have a lot of moving parts or complicated mathematical formulations that the BO community tends to favor these days, but I am in favor of acceptance for the reasons I mentioned above. It’s a rigorous scientific work that could have a significant impact on the way BO is performed in the HPO community today.

### Weaknesses:
My only criticism is that priors used (on continuous params) are Gaussian. The authors may be leaving performance on the table by using such a simple prior. But this does not affect my score that much.


## Additional Notes:
* The decay method you have was used in this workshop paper Cost-aware Bayesian optimization, by Lee et al., 2020 (in a slightly different context, to decrease the impact of a cost model), so you might consider citing that if you want to reinforce that this decaying type modification has precedence in the literature.
* There is a stronger connection here with metalearning, in the context of HPO, (which seeks to build and then exploit priors) as a whole, and perhaps this could be included as a short paragraph in the related work section w/ some additional citations (e.g., more of Hutter’s work).

**Summary Of The Paper:**

## Summary:

PiBO is a very straightforward paper that seeks to incorporate prior knowledge about the optimum to accelerate Bayesian optimization.

They achieve this by simply multiplying the acquisition function (Expected Improvement, in the paper) by the prior distribution, and then maximizing that. They then decay the prior in order to deal with mis-specified priors. The authors also provide some theory that this weighted acquisition function performs (for some particular definition of performance) no worse than EI times a constant that depends on the decay rate. It’s a reasonable idea; honestly I’m surprised no paper has tried this before.

Synthetic and real-world experiments indicate that:
-With a well-specified prior, PiBO demonstrates superior performance over it’s basic EI counterpart.
-With a poorly-specified prior, PiBO performs poorly worse than it’s EI counterpart but is usually able to recover given enough iterations.



**Summary Of The Review:**

The method is simple and sensible, and experiments convincingly demonstrate that it works with a well-specified prior, and more importantly, demonstrate that a poor prior could lead to problems. It’s a rigorous scientific work that could have a significant impact on the way BO is performed in the HPO community today.

---

> ### Author Response · Authors · 2021-11-12
> **Response to Reviewer eisp**
>
> We are glad that the reviewer agrees with our view on the importance of the topic and that the approach can have a significant impact on HPO.
>
> $~$
> >##### My only criticism is that priors used (on continuous params) are Gaussian. The authors may be leaving performance on the table by using such a simple prior.
>
> It is true that the performance over regular BO would likely improve by using a more tailored prior. However, we limited our experiments to Gaussian priors for two reasons:
>
>
>
> 1. It was the only distribution supported by all competing approaches (including the approaches in appendix F).
> 2. It made for the simplest procedure of constructing priors, least affected by our own biases.
>
> $~$
> >##### The decay method you have was used in this workshop paper Cost-aware Bayesian optimization, by Lee et al., 2020 (in a slightly different context, to decrease the impact of a cost model), so you might consider citing that if you want to reinforce that this decaying type modification has precedence in the literature. There is a stronger connection here with metalearning, in the context of HPO, (which seeks to build and then exploit priors) as a whole, and perhaps this could be included as a short paragraph in the related work section w/ some additional citations (e.g., more of Hutter’s work).
>
>
> We thank the reviewer for this feedback, and agree on the relevance of the stated references. We have included the Lee et al. paper in the context of decaying our prior, added an additional meta-learning reference, and point towards a survey about meta-learning for HPO to complement our paragraph on this topic (Section 2.3, Learning from Previous Experiments).

---

> > ### Author Response · Authors · 2021-11-13
> > **Additional response about use of different priors**
> >
> >
> >
> > We would like to expand on our reply to this comment by the reviewer:
> >
> >
> > >##### **My only criticism is that priors used (on continuous params) are Gaussian. The authors may be leaving performance on the table by using such a simple prior.**
> >
> > While we justified the use of only Gaussian priors in our experiments in our previous post (the code for the baselines only supports Gaussian priors and Gaussians are the simplest choice, likely least affected by our own biases), we do agree with the reviewer that it would be useful to also show an experiment with non-Gaussian priors. We therefore just conducted a new experiment with non-Gaussian priors that we would like to report.
> >
> >
> > Firstly, we acknowledge the subjective nature of this experiment. We set the priors based on our own experience and the nature of the tasks at hand, without knowledge of which configurations yielded our best results; others might have chosen different priors and obtained different results, but this is the only experiment that we performed (we did not cherry-pick).  The setup we used for this experiment was the same as for Figure 3 in the paper: optimizing five hyperparameters of MLPs for six OpenML datasets. The hyperparameters and their predetermined ranges and their priors are outlined below, as well as our reasoning for selecting these priors.
> >
> > | Parameter      | Range | Prior |
> > | ----------- | ----------- | ----------- |
> > | Network depth      | {${1, 2, 3}$}        | {${0.01, 0.09, 0.9}$}       |
> > | Network width   | $[2^4. 2^{10}]$       | $\text{Beta}(3,2)$       |
> > | L2 Regularization   | $[10^{-8}, 1]$       | $\text{Beta}(3,2)$       |
> > | Batch size   | $[2^2. 2^{8}]$       | $\text{Beta}(2,2)$       |
> > | Initial learning rate   | $[10^{-5}, 1]$       | $\text{Beta}(2,5)$       |
> >
> >
> >
> > Identical priors were used for all six datasets. We chose beta priors for all continuous parameters due to their closed support and the ease with which they can be designed. Since the lower ranges of the hyperparameters for network depth and width would yield very small neural networks we opted to put more probability mass on the larger values for both depth and width. To balance this, we set the L2 regularizer to a moderately high value. We biased the learning rate to have high density on the typical $10^{-4}$ to $10^{-3}$ range. For the batch size, we simply centered the prior in the search space, as we did not know a priori whether the tasks required particularly small or large batch sizes.
> >
> >
> > We provide the results in this figure: [https://imgur.com/a/inNOqmv](https://imgur.com/a/inNOqmv)
> >
> > The figure shows the performance of $\pi$BO with the previous Gaussian prior centered on defaults, $\pi$BO with Beta priors set as discussed above, and sampling from the respective priors.
> >
> > As evidenced by the figure, this beta prior yielded substantially better results for $\pi$BO than our previous Gaussian prior on 2 out of 6 tasks (Australian Credit and German Credit), with very similar results for the other 4 datasets. $\pi$BO also improved over sampling from the prior for all six datasets. We believe that this experiment demonstrates that $\pi$BO can be used effectively with non-Gaussian priors, and that doing so indeed can improve over Gaussian priors only. Thanks a lot to the reviewer for suggesting this, as we do believe it makes our paper stronger.

---

> > > ### Comment · Reviewer_eisp · 2021-11-29
> > > **Reply to additional response**
> > >
> > > Thank you for running this additional experiment! I'm glad the results are good!
> > >
> > > My score remains unchanged (it was an accept to begin with), but I deeply appreciate the effort shown to reply to reviewer concerns, regardless of their score.

---

### Official Review · Reviewer_VqSx · 2021-11-02

**Correctness:** 2
**Technical Novelty And Significance:** 2
**Empirical Novelty And Significance:** 2
**Recommendation:** 6
**Confidence:** 4

**Main Review:**

Overall this paper is well-written and appropriately discusses the related literature.

While the problem of incorporating the optimum location makes sense, I am not sure if the experts can provide a proper probability distribution about the optimum. Finding this information in practice seems rare. For example, even the experiments in section 4.4 rely on the priors which are based on already (manually) tuned deep networks. Finding useful priors may thus be hard in practice.

The method seems simple and as stated in the paper is easy to implement. However, it seems to me that while the paper claims the proposed method to work with any acquisition function, it gives analysis only for EI acquisition function? Is this true?

Also, there is a claim that the method can recover from any misleading prior, but it may be true only when there is a nonzero support for the true optimum x*. If the prior likelihood for x* is zero, the method may not converge. In case of misleading prior with nonzero support, is the convergence rate still sublinear?

In Eq 4, what is y^*_{n+1}? It seems that left-hand side of the EI expression in Eq 4 does not depend on x, which is strange.

In the experiments section, all the methods seem to be starting from different initialization points, if this is true then the comparison between the methods may not be fair. Why does \pi-BO often start from a higher starting value after the initial design? We do not know if \pi-BO does well due to the proposed algorithm or due to the better initialization?


Minor Comments:
On page 1: “documentesd” should be “documented”
On page 4: in the third para, “show” should be “shows”
On page 5: before Eq 6, “prior-weighed” should be “prior-weighted”

**Summary Of The Paper:**

This paper proposes a new method to incorporate prior knowledge about optima location in Bayesian optimization. The paper claims that the proposed method is simple to implement, can be used with any acquisition functions and has efficient convergence rate.

**Summary Of The Review:**

While the proposed method for this problem makes sense, it may be hard for experts to provide such information in practice. Further, the convergence analysis seems limited to Expected improvement (EI) acquisition function only. Last but very important, different methods seem to be starting from different initialization points, which may be problematic.

---

> ### Author Response · Authors · 2021-11-12
> **Response to Reviewer VqSx**
>
> We thank the reviewer for the thorough feedback and for appreciating the simplicity of our approach. We have addressed your request for runs with identical initialization and your minor remarks in the new version of the paper, and now reply to the main review.
>
> $~$
> >##### While the problem of incorporating the optimum location makes sense, I am not sure if the experts can provide a proper probability distribution about the optimum. Finding this information in practice seems rare. For example, even the experiments in section 4.4 rely on the priors which are based on already (manually) tuned deep networks. Finding useful priors may thus be hard in practice.
>
> The interviews conducted with practitioners in Wang et. al. (2019), and the recommendations on hyperparameter ranges by Smith (2018), indicate that expert knowledge exists. Additionally, the prevalence of manual tuning indicates that practitioners trust their ability to tune hyperparameters themselves. In Section 4.4, we displayed, without including our own biases, that the gap from suggesting a configuration (such as a library default, or manual search) to a well-constructed prior over the optimum need not be large. One simply needs one, or a small number of suggested good points, and construct distributions around those points, to obtain a prior that reflects one’s belief reasonably well. Considering the prevalence of manual tuning and library defaults, we believe that coming up with typical hyperparameter values to construct a prior around should be doable and is promising in practice.
>
> $~$
> >##### The method seems simple and as stated in the paper is easy to implement. However, it seems to me that while the paper claims the proposed method to work with any acquisition function, it gives analysis only for EI acquisition function? Is this true?
>
> We empirically demonstrate the effectiveness of $\pi$BO with the UCB, TS and PI acquisition functions (see Figure 7 and Appendix B).  However, we only provide convergence analysis for EI.
>
> $~$
> >##### In the experiments section, all the methods seem to be starting from different initialization points, if this is true then the comparison between the methods may not be fair. Why does $\pi$BO often start from a higher starting value after the initial design? We do not know if $\pi$BO does well due to the proposed algorithm or due to the better initialization?
>
> BOPrO, BOWS and $\pi$BO indeed have different initializations. We followed the guidelines of the original implementations by the authors to run BOPrO and BOWS. While $\pi$BO retrieves the mode of the prior (if available) as its first design point, BOPrO simply samples from it and BOWS runs regular BO in the warped space. However, we agree that a comparison under the same initializations is very useful. Below, we provide runs on the MLP tasks where BOPrO and BOWS share $\pi$BO’s initialization, evaluating the mode as the first point. We see that πBO offers the best performance on four out of six tasks,and displays the most consistent performance across tasks. We have now integrated these results in the paper and adjusted the text accordingly. Redoing the experiments from Section 4.4 with identical initializations requires more time and computational resources, but we have started them and will integrate them in the paper as soon as they are finished.
>
> [https://imgur.com/a/PMT2Dbe](https://imgur.com/a/PMT2Dbe)
>
> $~$
> >##### Also, there is a claim that the method can recover from any misleading prior, but it may be true only when there is a nonzero support for the true optimum x*. If the prior likelihood for x* is zero, the method may not converge. In case of misleading prior with nonzero support, is the convergence rate still sublinear?
>
> It is true that the method is not guaranteed to converge if we allow for zero-valued support. Because of that, in Section 2.1, we state:
>
> “We note that, without loss of generality, we require π to be strictly positive on all of X, i.e., any point in the search space might be an optimum.” There is both a practical and a theoretical reason for this, and you have already pointed out the theoretical one. The practical reason is rather simple: If the user genuinely believes that a point has exactly zero probability of being the optimum, it should be excluded from the search space altogether.
>
> However, we guard against the user specifying a prior with zero support somewhere (as it would likely not be their intention - e.g. through rounding errors) by adding a small probability of $\epsilon = 10^{-12}$ for all parts of the search space and renormalizing (this information is admittedly fairly hidden, in Appendix F). As such, $\pi$BO will still converge under this prior.
>
>
> We thank you again for your thorough review. We hope that with our answers we could clarify your concerns and, if so, we would appreciate it if you would consider increasing your score. If you have any further questions or comments please do not hesitate to post.

---

> > ### Author Response · Authors · 2021-11-19
> > **Additional response to Reviewer VqSx**
> >
> > We once again thank the reviewer for the initial feedback. We would love to hear from the reviewer if our previous answer addressed the reviewer's concerns. These are the comments that have been addressed:
> > - Experimental setup and initialization, with adjusted results in Section 4.3 as a consequence
> > - Clarification of the convergence of our approach for zero-valued priors
> >
> > We hope that we have alleviated some of your concerns regarding our work, and if not, we would be happy to consider any additional feedback you may have and discuss further.

---

> > > ### Author Response · Authors · 2021-11-22
> > > **Additional response to Reviewer VqSx**
> > >
> > > In re-reading the reviewer’s comments we noticed that we had missed one question:
> > >
> > > > ##### “In Eq 4, what is $y^*_{n+1}?$ It seems that left-hand side of the EI expression in Eq 4 does not depend on x, which is strange.”
> > >
> > > We agree that this equation was imprecise, and have adjusted the text accordingly. The updated Eq. 4 can also be found here: https://imgur.com/a/RZuvxYS

---

> > > > ### Comment · Reviewer_VqSx · 2021-11-29
> > > > **Post rebuttal feedback**
> > > >
> > > > I have read the authors rebuttal and my worries about the different initialisation across different methods has reduced. I am increasing my score to 6 now. I still have a minor concern on the initialisation process: why is the initialisation not using same random seed across the methods? By keeping the same seed, we are giving each method the same starting point.
> > > > Another concern is that the convergence analysis is limited only to EI and it would be great to extend this to other acquisition functions.

---

> > > > > ### Author Response · Authors · 2021-11-30
> > > > > **Response to Post rebuttal feedback**
> > > > >
> > > > > We thank the reviewer for the additional response, and for the adjustment in the score. We agree that keeping the initialization identical will improve the comparability of the methods. As such, we provide new results for the HPOBench benchmarks below with fixed, identical seeds. We also started equivalent runs for the DL experiments; while these will not finish before the end of the rebuttal, we will add their results to the camera-ready version.
> > > > >
> > > > > https://imgur.com/a/yot6Zrb

---

### Official Review · Reviewer_gZca · 2021-11-03

**Correctness:** 4
**Technical Novelty And Significance:** 3
**Empirical Novelty And Significance:** 3
**Recommendation:** 8
**Confidence:** 5

**Main Review:**

I am quite aware of research in BO and I think integrating prior information is an important aspect that has not been looked into much. Existing solutions are either restricted or do not admit a convergence analysis. In that respect, I quite like the idea of the paper. It's quite simple and yet lend itself to an analysis of the convergence. The experiments are sufficient in my opinion to expose the behaviour of the algorithm at different scenarios.

Theoretical analysis is convincing. Although I should say that the upper bound can become uselessly loose if the prior is not designed properly. The constant C_{\pi, n} can be arbitrarily large if the prior is Gaussian like and narrow. This may make the anytime upper bound not useful. The authors should use this insight to restrict the shape of the prior function. Another limitation of the theoretical analysis is sticking to EI acquisition function. An analysis that admit GP-UCB acquisition function would have completed the work because unfortunately, EI is not proven to converge for noisy function case.

Another important question is how do the authors propose to scale the prior function \pi so that can influence the acquisition function. The EI functions are notoriusly peaky in the sense that it can have sharp peaks among the valley of mostly small values. In that case, to make prior count, it needs to be scaled properly. I would like the authors comment on this aspect.

**Summary Of The Paper:**

The paper proposes a method to incorporate prior information about the optimum in the standard Bayesian optimisation (BO) setting. The prior is specified as a smooth function over the range having high values where the experimenter think the optimum may exist with high probability and low values otherwise. This prior function is incorporated in the BO workflow through multiplication with the EI acquisition function. The authors show that when the prior function is decayed then this BO achieves the usual sublinear regret rate of a standard BO asymptotically. The synthetic experiments are extensive and convincing. The real case studies are limited, but sufficient.

**Summary Of The Review:**

The paper proposes a simple but elegant method to incorporate prior information about optimum in the BO workflow. It's a long-standing research gap and the proposed solution fills the gap to some degree. The best part is that the simple way of integrating prior makes its analysis straightforward using the current techniques. However, it has got some limitations: 1) lack of extension to GP-UCB acquisition function, 2) lack of discussion around the extreme looseness in the bound that can happen when the prior is chosen improperly, and 3) lack of discussion around the relative scaling between the acquisition function and the prior function and its impact on the influence of the prior function in the process. I would go for weak accept. But I am ready to move up depending on the rebuttal.

---

> ### Author Response · Authors · 2021-11-12
> **Response to Reviewer gZca**
>
> We are glad that the reviewer appreciates the relevance of the topic as well as the utility of our method. We also thank the reviewer for the thorough feedback.
>
> $~$
> > ##### **Theoretical analysis is convincing. Although I should say that the upper bound can become uselessly loose if the prior is not designed properly. The constant $C_{\pi, n}$ can be arbitrarily large if the prior is Gaussian like and narrow. This may make the anytime upper bound not useful. The authors should use this insight to restrict the shape of the prior function.**
>
> The reviewer is correct in that the constant $C_{\pi, n}$ can be small if the prior is too confidently designed. Moreover, we agree that $C_{\pi, n}$ can be used for additional insight. We have added an additional section in the appendix on how various prior strengths affect the constant $C_{\pi, n}$ for a given iteration, to highlight how the prior strength affects the upper bound on convergence. This plot is also linked here. This new appendix section can educate the user's decision when constructing the prior by displaying the convergence one is expected to get, relative to that of EI. We see that, for approximately half of the space, the upper bound on the loss is at least 80% (bright green or yellow) of the upper bound of EI, and only a small region of very narrow priors (dark blue) give a low guaranteed convergence rate.
>
> **[https://imgur.com/a/jTYxXx5](https://imgur.com/a/jTYxXx5)**
>
> The figure is showing the value of the constant $C_{\pi, n}$ at iteration 50 for different values of $\beta$ and $\sigma$ in the setting where $\pi$ is a one-dimensional Gaussian distribution with mean in the center of the search space. Regions in dark represent pairs of ($\sigma, \beta$) where the upper bound on prior-weighted EI is low relative to EI, whereas regions in yellow represent pairs of ($\sigma, \beta$) where the upper bound is approximately the same.
>
> Empirically, the “Wrong” prior in the synthetic section of our results covers an extreme case of the scenario the reviewer is describing, where $\sigma$ is only $1\\%$ of the length of the search space, and located far away from the optimum. While the first iterations of $\pi$BO are evidently weak, it recovers to a level where it is still competitive with regular BO within a few tens of iterations.
>
> $~$
> >##### Another limitation of the theoretical analysis is sticking to EI acquisition function. An analysis that admit GP-UCB acquisition function would have completed the work because unfortunately, EI is not proven to converge for noisy function case.
>
> While we agree with the reviewer that bounds for additional acquisition functions would indeed be useful, we are unable to provide such bounds at this stage. However, we would like to mention that, as shown by Nguyen et. al. (2017), EI does converge in a setting with sub-gaussian noise.
>
> Vu Nguyen, Sunil Gupta, Santu Rana, Cheng Li, Svetha Venkatesh. Regret for Expected Improvement over the Best-Observed Value and Stopping Condition. _Proceedings of the Ninth Asian Conference on Machine Learning_, PMLR 77:279-294, 2017.
>
>
>
> $~$
> >##### Another important question is how do the authors propose to scale the prior function $\pi$ so that can influence the acquisition function. The EI functions are notoriusly peaky in the sense that it can have sharp peaks among the valley of mostly small values.
>
> We note that since we multiply (and not add) the acquisition function and the prior function, both don’t have to live on the same scale. Acquisition functions can indeed often be multimodal, have a small number of sharp peaks, and be almost zero everywhere else. As displayed for the well-located prior in Figure 1, $\pi$BO **influences the selection between peaks** at both iteration 4 and iteration 6. Only at iteration 8, does it make a compromising selection: one that is moderately good under both the prior and EI. We believe that the **selection between peaks** shows that the prior affects decision-making, even in the case where the acquisition function is peaked the way the reviewer is describing.
>
> **Clarifications:**
>
>
> * The prior need not necessarily be smooth. In fact, the deep learning experiments contain categorical variables, for which we specify discrete priors. This is outlined in full in Appendix F.3.
>
> We hope that our answers clarified your concerns. You mentioned that you might move your score up depending on the rebuttal, and we would very much appreciate it if you did. If you have any further questions or comments please do not hesitate to post.

---

> > ### Comment · Reviewer_gZca · 2021-11-17
> > **Thank you for the clarifications**
> >
> > Thank you for the clarifications. These are my reflections on your rebuttal:
> >
> > 1. While those plots are okay, I think a better message would be to convey the caveat against choosing a prior with unnecessary high confidence.
> >
> > 2. I am aware of that work, and I think there are still interesting missing pieces in the puzzle. But that is a side point. I think it is a matter of time before someone comes with a proof of EI for noisy observations. Until then the analysis remains incomplete.
> >
> > 3. I am convinced by your clarification.
> >
> > As promised, I will revise my review score upward.

---

### Author Response · Authors · 2021-11-12
**Changes to the paper**

First of all, we once again want to thank the reviewers for their effort. We hope to continue the constructive, fruitful discussion in the coming weeks.

After considering the feedback given to us, we have made the following changes to the paper:
- Adhering to the request of Reviewer VqSx, BOPrO and BOWS now share $\pi$BO's initialization on the HPOBench MLP tasks. The updated results for these tasks can be found in Section 4.3.
- Added appendix section E.1, "Sensitivity analysis on $C_{\pi, n}$", to provide insight into the interplay between $\beta$, $\sigma$ and convergence rate
- Integration of minor remarks

---

> ### Author Response · Authors · 2021-11-21
> **Additional changes to the paper**
>
> We would like to let the reviewers know that we have made one additional change to the PDF. We have updated the plots on Imagenette-128 and U-Net Medical (Section 4.4) to address the request of reviewer VqSx. Now, BOWS, BOPrO and Prior Sampling share $\pi$BO's initialization, meaning that they all pick the mode of the prior as their first design point. While the performances on the first iteration are still not identical, the remaining difference is solely due to objective function noise. With these changes, the BOPrO and BOWS produce slightly better results on ImageNette, but the ranking of approaches remains the same. For U-Net Medical, there are no substantial differences compared to previous results.
>
> The new results can also be found here:
> https://imgur.com/a/ZyWKDhZ

---

### Decision · Program_Chairs · 2022-01-20

**Decision:**

Accept (Poster)

**Comment:**

This paper investigates Bayesian optimization where a prior distribution over the optimal is available. The authors conducted a systematic study on a very intuitive prior-augmented acquisition function that multiplication the prior probability with the EI heuristic --- including an asymptotic analysis on the regret, comprehensive (controlled) synthetic experiments, and moderate empirical support on several real-world case studies.

All reviewers find the paper well-written and appreciate the rigor of the empirical evaluation. The theoretical analysis is also helpful to provide additional justification for the proposed approaches. I would like to add that the paper also included a brief but comprehensive survey on prior work related to leveraging prior in BO, which I find useful for the general audience.

Reviewers noted that such a bound could become trivial with a bad prior in practice, and further suggest that one may leverage these theoretical insights as general guidelines to practitioners in designing the prior. I think this is a valuable message to convey and suggest authors take it into account in the revision.

There were initial confusions pertaining to the experimental details, mainly concerning the effect of the quality of the prior on the performance of the proposed algorithm. The authors provided an effective rebuttal with much concrete empirical support, and after a few rounds of interaction during the discussion phase, the reviewers are convinced about the empirical significance of the proposed work. Overall, this makes a solid work.